# Structure and catalytic regulation of *Plasmodium falciparum* IMP specific nucleotidase

Loïc Carrique [1,3,7], Lionel Ballut [1,7], Arpit Shukla [2,4,7], Neelakshi Varma [2], Resmi Ravi [2], Sébastien Violot [1], Bharath Srinivasan [2,5], Umesh Tippagondanahalli Ganeshappa [2,6], Sonia Kulkarni [2], Hemalatha Balaram [2✉] & Nushin Aghajari [1✉]

*Plasmodium falciparum* (*Pf*) relies solely on the salvage pathway for its purine nucleotide requirements, making this pathway indispensable to the parasite. Purine nucleotide levels are regulated by anabolic processes and by nucleotidases that hydrolyse these metabolites into nucleosides. Certain apicomplexan parasites, including *Pf*, have an IMP-specific-nucleotidase 1 (ISN1). Here we show, by comprehensive substrate screening, that *Pf*ISN1 catalyzes the dephosphorylation of inosine monophosphate (IMP) and is allosterically activated by ATP. Crystal structures of tetrameric *Pf*ISN1 reveal complex rearrangements of domain organization tightly associated with catalysis. Immunofluorescence microscopy and expression of GFP-fused protein indicate cytosolic localization of *Pf*ISN1 and expression in asexual and gametocyte stages of the parasite. With earlier evidence on *isn1* upregulation in female gametocytes, the structures reported in this study may contribute to initiate the design for possible transmission-blocking agents.

[1] Molecular Microbiology and Structural Biochemistry, UMR5086 CNRS-University of Lyon, Lyon 69007, France. [2] Molecular Biology and Genetics Unit, Jawaharlal Nehru Centre for Advanced Scientific Research (JNCASR), Bengaluru, Karnataka 560064, India. [3] Present address: Division of Structural Biology, Henry Wellcome Building for Genomic Medicine, University of Oxford, Oxford OX3 7BN, UK. [4] Present address: Venture Studio, Ahmedabad University, Navrangpura, Ahmedabad 380009, India. [5] Present address: Mechanistic Biology and Profiling, Discovery Sciences, R&D, AstraZeneca, Cambridge, UK. [6] Present address: Rubizon Pvt ltd, #783, 10th cross, Mathrushree, Bangalore 560086, India. [7] These authors contributed equally: Loïc Carrique, Lionel Ballut, Arpit Shukla. ✉email: hb@jncasr.ac.in; nushin.aghajari@ibcp.fr

*P*lasmodium falciparum (*Pf*), causing the most severe form of malaria, is a purine auxotroph salvaging nucleobases and nucleosides directly from the host to ensure its development. Indispensable for parasite survival, the purine salvage pathway constitutes an important target for the development of new antimalarial drugs[1–3]. Purine nucleotides are mandatory for various cellular processes, including DNA and RNA synthesis, as energy metabolites (ATP, GTP) and for cofactor (NAD, FAD, FMN, CoA) synthesis. During the intraerythrocytic stages, adenosine and hypoxanthine salvaged from the human host are converted into inosine monophosphate (IMP), a precursor for the synthesis of both adenosine monophosphate (AMP) and guanosine monophosphate (GMP)[2]. Maintaining the optimal nucleotide balance required for normal cell proliferation is controlled by 5′-nucleotidases amongst other processes[4]. These enzymes catalyze the dephosphorylation of ribo- and deoxyribonucleoside monophosphates to their corresponding nucleosides, and also exhibit, in certain cases phosphotransferase activity[5]. Whereas seven 5′-nucleotidases are characterized in humans[6,7], only one is annotated in *Plasmodium* species[8]. Classified as an IMP-specific 5′-nucleotidase (ISN1, EC 3.1.3.99) based on sequence homology with a yeast counterpart[9], this enzyme possesses the four characteristic Haloacid Dehalogenase (HAD) superfamily motifs[10] (Supplementary Fig. 1A), but no significant sequence similarity with human 5′-nucleotidases. Unlike the well-studied cytosolic nucleotidase-II (cN-II) class of human purine 5′-nucleotidases, structure–function relationship studies on ISN1 are lacking. *Plasmodia* lack the cN-II class of purine 5′-nucleotidases that are present in many prokaryotes and eukaryotes including humans.

Recently, Brancucci et al. revealed the ability of *P. falciparum* to sense and process host-derived physiological signals, and demonstrated that lysophosphatidylcholine (LysoPC) present in the host serum represses sexual differentiation in the parasite[11]. Gametocyte formation brought about by LysoPC depletion is associated with activation of expression of more than 300 genes, including genes involved in phosphocholine (PC) biosynthesis, DNA replication and macromolecule modification. Interestingly, ISN1 is also strongly induced. However, genome-wide disruption in *P. falciparum* by piggyBac transposon insertion suggests that ISN1 is mutable in the asexual stages without loss of fitness[12]. To elucidate the biochemical and the physiological functions of *P. falciparum* ISN1 (*Pf*ISN1), we here report the substrate specificity determination along with the kinetic parameters of the enzyme. We moreover report the crystal structures of a member of this enzyme sub-family, namely that of the apoenzyme, variants and complexes with various ligands adopting both open and closed conformations. It should be noted that prior to this report, a structure of the member of ISN1 family is not reported. We also show the cytosolic localization in asexual and sexual stages of the parasite. These structural and functional studies reveal the molecular reaction mechanism employed in this family of 5′-nucleotidases and their regulation via inter-domain interactions.

## Results

### Expression and localization of ISN1 in *Plasmodium*. Among *Plasmodia*, only species infecting humans, primates and birds have the *ISN1* gene (Supplementary Fig. 1B) whereas the rodent parasite species lack a homologous sequence. Plasmodial ISN1 sequences are similar in length and exhibit 82–100% sequence identity. The 444 amino acids *Pf*ISN1 encoded by the *PF3D7_1206100* gene contains nine exons and eight introns. Apart from the conserved synteny, the intron–exon boundaries are also fully conserved, suggesting a gene loss/gain event during evolution of different species of *Plasmodia*.

Reverse transcriptase-polymerase chain reaction of the RNA isolated from the trophozoite stage of intraerythrocytic parasites showed the presence of a single-spliced product of about 1350 base pairs (Supplementary Fig. 1C), thereby confirming the expression of the gene in the intraerythrocytic asexual stages and subsequent sequencing validated the predicted splice junctions. Indirect immunofluorescence microscopy using anti-*Pf*ISN1 antibodies showed cytoplasmic localization in the intraerythrocytic asexual and sexual stages of *P. falciparum* (Fig. 1a and Supplementary Fig. 2). Moreover, live-cell imaging of *P. falciparum* and *P. berghei* parasites episomally expressing *Pf*ISN1 fused to GFP confirmed the cytoplasmic localization (Fig. 1b, c).

### Overall fold. *Pf*ISN1 crystal structures were determined for the active enzyme in Apo-, ΔC10-truncated, ΔN59-truncated, ATP-bound forms, and for the inactive mutant D172N (full-length and ΔN30-truncated) bound to IMP and Mg$^{2+}$ (Supplementary Table 1). All structures adopt a tetrameric assembly displaying a cross-like shape (Fig. 2a), in agreement with the quaternary solution structure (Supplementary Table 2, Supplementary Figs. 3 and 4). Each subunit (Fig. 2a, b) consists of four domains: an "N-Terminal Regulatory Domain" (NTRD, M1 to K59), an "OD" (D60 to T143), a HAD-like Catalytic Domain (CD, F144 to K270 and K371 to Q444) and a C2 cap domain[13] (K271 to N370) (Fig. 2b, e).

As in other 5′-nucleotidases[14], the catalytic domain possesses the four motifs characteristic of the HAD phosphatase family and an α/β Rossmann-like fold with a seven-parallel stranded β-sheet surrounded by eight α-helices (α7 to α12, α15 and α16) (Fig. 2b). The C2 cap domain forms the core β-sheet (four anti-parallel β-strands) carrying W365 involved in IMP stacking and two α-helices (α13 and α14) interacting with the oligomerization domain (OD) of another subunit to form "dimer interface 1" (Fig. 2c). Furthermore, helices α3 to α5 from one OD interact with OD of another subunit to form "dimer interface 2" (Fig. 2d). Finally, the NTRD is mainly composed of a random coil, exempting helices α1 and α2, both situated at interface 1 between α and β subunits (or between γ and δ subunits, Fig. 2c) in the apo conformation (*Pf*ISN1-Apo). However, the cavity (Fig. 2c) created by the interaction between the two subunits at interface 1 can accommodate only one NTRD at a time, forcing the second NTRD to localize elsewhere. No extra density is observed for this latter NTRD up to residue 59, thereby suggesting it being highly flexible. When superimposing subunits with and without NTRD, the overall structures are very similar (RMSD 0.4 Å). Analyses of *Pf*ISN1-Apo SAXS data confirm the high flexibility of two NTRDs, and SAXS restrained modelling indicates that unstructured NTRDs may be located at interface 2 (Supplementary Fig. 5).

### Biochemical characterization and substrate specificity. HAD superfamily phosphatases exhibit broad substrate specificity[15–17]. In order to gain insight into the physiological function of *Pf*ISN1, 61 possible substrates were screened (Supplementary Table 3). At a concentration of 10 mM, only IMP and, to a much lower extent, AMP and p-nitrophenyl phosphate (pNPP, non-physiological substrate) were found to be substrates with specific activity values of $3.2 \pm 0.5$, $0.07 \pm 0.03$ and $0.002 \pm 0.0002\ \mathrm{s}^{-1}$, respectively. *Pf*ISN1 displayed maximum activity in the presence of Mg$^{2+}$ as cofactor and over the pH range 4.0–5.0 (Supplementary Fig. 6A). For IMP, the enzyme turnover, $k_{\mathrm{cat(app)}}$ remained unchanged within pH 4.0–9.0, whereas the apparent $K_{\mathrm{m}}$ value dropped 194-fold upon change in pH from 8.0 to 5.0 (Table 1, Supplementary Fig. 6B, C). IMP has three ionizable groups with p$K_{\mathrm{a}}$ values of 1.5, 5.8 and 9.1[18,19] where the first two values correspond to

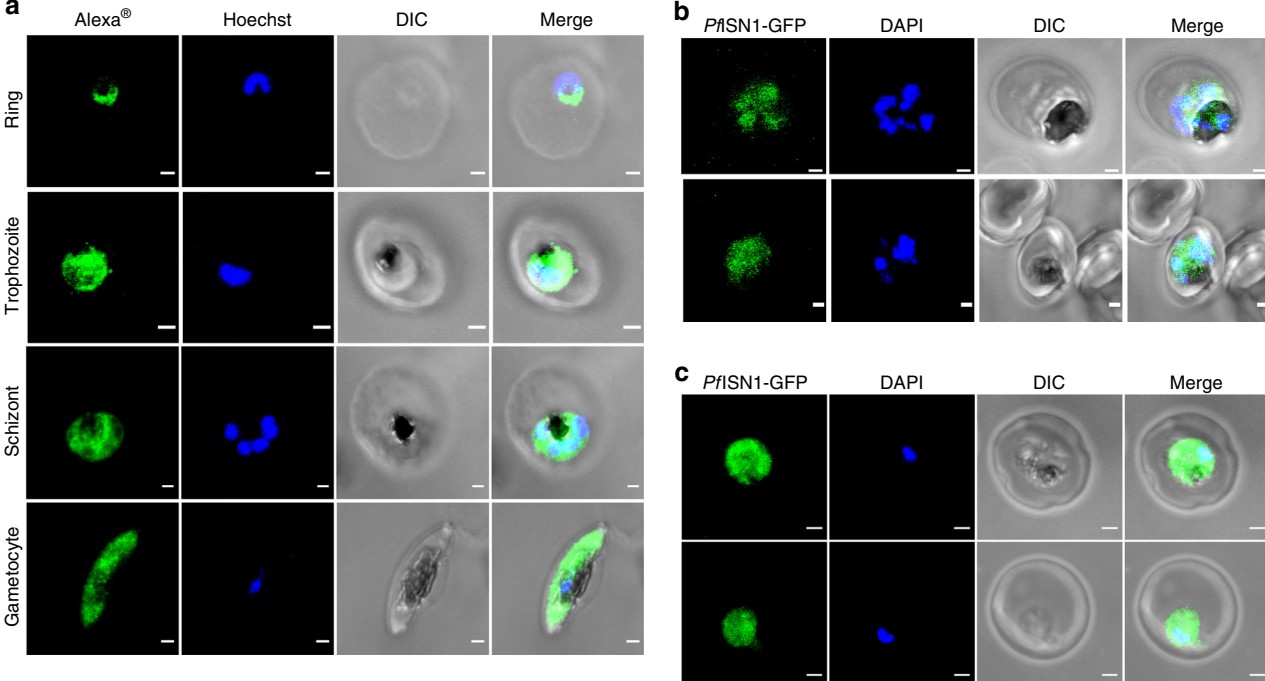

**Fig. 1 Localization of ISN1 in *Plasmodium*. a** Indirect immunofluorescence microscopy with anti-*Pf*ISN1 antibody on intraerythrocytic asexual (3D7 strain) and sexual (3D7A strain) stages of *P. falciparum*. The absence of signal from pre-immune control ruled out non-specific binding (Supplementary Fig. 2). The antibody used was generated as described in "Methods". Gametocytes were enriched as described in "Methods". The experiment was repeated at least three times. Live-cell fluorescence imaging of episomally expressed *Pf*ISN1-GFP fusion protein in 3D7_*Pf*ISN1-GFP (**b**) and *Pb*ANKA_*Pf*ISN1-GFP (**c**) parasites. Imaging of *Pb*ANKA_*Pf*ISN1-GFP was performed at least three times with parasites collected from independent mice, and imaging of 3D7_*Pf*ISN1-GFP was performed at least three times with parasites harvested from independent cultures. Scalebars correspond to 1 μm. The stages indicated correspond only to (**a**). Source data are provided as a Source data file.

ionization of the –OH groups on the phosphate, and the third value corresponds to the –N₁H of the purine. The drop in the apparent $K_m$ suggests that the mono-anionic species with a single negative charge on the phosphate moiety is the form of the substrate that binds to the enzyme. The apparent $pK_1$ and $pK_2$ values of $4.95 \pm 0.32$ and $6.24 \pm 0.36$, respectively (Supplementary Fig. 6B) might reflect that both substrate ionization and an acid group on the enzyme are involved in binding.

Substrate/cofactor saturation plots varied as a function of pH and substrates/cofactor. Plots for IMP were hyperbolic at both pH 8.0 and 5.0, AMP and $Mg^{2+}$ switched from hyperbolic at low pH to sigmoidal at pH 8.0 and pNPP was sigmoidal at pH 8.0 (Supplementary Fig. 6C–F), and hence suggest cooperativity across subunits in the tetramer. At low pH, IMP was the preferred substrate with maximum catalytic efficiency (Table 1). AMP, on the other hand, remains a poor substrate at low pH. At pH 8.0, the enzyme shows 466-fold lower catalytic efficiency on pNPP (though a non-physiological substrate) than on IMP (Table 1). *Pf*ISN1 did not show phosphotransferase activity with IMP-adenosine as phosphate donor-acceptor pair unlike other purine 5′-nucleotidases that display this catalytic feature[5] (Supplementary Fig. 8).

**IMP-binding site**. The two invariant aspartyl residues in Motif I of HAD superfamily enzymes are involved in catalysis with the first aspartate (D170) being the phosphate acceptor and the second (D172) coordinating $Mg^{2+}$[20]. Mutants D170N, D172N, D172A and D170N-D172N were inactive on IMP, both at pH 8.0 and 5.0 (Fig. 3). Surprisingly, on the synthetic compound pNPP, D172N and D172A mutants showed 91- and 15-fold increased activity, respectively (Supplementary Fig. 9A), and hence could be

used to examine IMP binding through inhibition of pNPP hydrolysis. Due to the highest binding affinity of D172N for IMP ($K_{d(app)}$ of $4.5 \pm 0.3$ μM vs. $83.3 \pm 3.9$ μM for D172A, (Supplementary Fig. 9B) and structural similarity, this mutant was selected for structure determination of ligand-bound *Pf*ISN1.

The crystal structures (Fig. 4a, b) confirm that the IMP-$Mg^{2+}$ substrate is mainly coordinated by conserved residues from the four HAD motifs, and by W365, which is situated on β-strand I. The base moiety of the nucleotide is stabilized by a π-stacking interaction with W365 and by three hydrogen bonds with A205 (Motif II), S207 and D367. Hydroxyl groups 2′ and 3′ of the sugar moiety form hydrogen bonds with D363 and D178, respectively (Fig. 4a), whereas the phosphate moiety of the nucleotide forms hydrogen bonds with D170 (Motif I), T204 and A205 (Motif II), K371 (Motif III), and N401 (Motif IV) (Fig. 4b). $Mg^{2+}$ is coordinated by the phosphate moiety, side chains of D170 (Motif I), D394 and Q395 (Motif IV), and the main chain carbonyl of D/N172. Finally, D402 (Motif IV) completes this organization by stabilizing the K371 side chain via a salt bridge. Motif IV mutants D394V, D402V and mutants D363V, D367V and W365L were inactive on IMP at both pHs, whereas activities of W365Y and W365F were comparable to the wild-type enzyme (Fig. 3). This infers that the aromatic nature of residue 365 is critical for IMP binding.

In addition, three highly conserved residues, Y176, R218 and D178 (Supplementary Fig. 1A) from the immediate vicinity of D172, may participate in the correct orientation of the catalytic residue during catalysis (N172 in Fig. 4b). Indeed, R218L was inactive or highly impaired at both pH values, Y176L showed significant activity loss whereas D178V was less affected (Fig. 3).

The activity of purine 5′-nucleotidases is modulated by the binding of substrate and effector molecules such as ATP, GTP

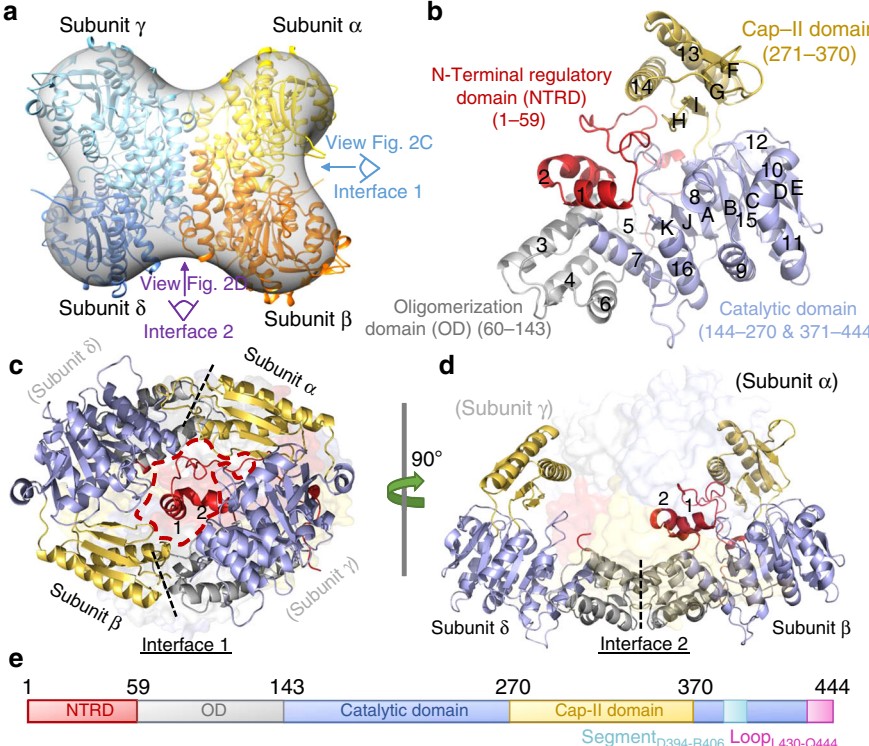

**Fig. 2 Overall structure of the *Pf*ISN1 apoenzyme. a** Crystal structure of the tetrameric *Pf*ISN1-Apo form. The structure has been fitted within the 25 Å resolution envelope calculated after single particle reconstruction from negatively stained samples. Each subunit constituting the homo-tetramer displays different colour coding. The orientation as seen in Fig. 2c, d are indicated. **b** A subunit (monomer) of *Pf*ISN1 with the N-terminal regulatory domain (NTRD) in red, the oligomerization domain (OD) in grey, the catalytic domain in blue and the cap domain in yellow. α-Helices are numbered from 1 to 16 and β-strands from A to K. **c** Dimer interface 1. Dashed lines indicate the dimer interfaces formed by the interaction between the OD of one subunit and the cap domain of another subunit. Subunits γ and δ (light grey) reside behind subunits β (left-hand side) and α (right-hand side), respectively. The red dashed contour delimits the cavity which accommodates an NTRD from subunit α. **d** Dimer interface 2. The dashed line indicates the dimer interface formed by the interaction between two OD's of subunits β and δ. Subunits in parentheses are depicted as surfaces while subunits in light grey are situated in the background (not visible in the figure). **e** Schematic linear illustration of *Pf*ISN1, with the colour code used in other panels, and with Segment$_{D394-R406}$ in cyan and Loop$_{L430-Q444}$ in pink.

**Table 1 Kinetic parameters and catalytic efficiency of *Pf*ISN1 wild-type and mutants[a].**

| Protein | Substrate (modulator) | pH | $k_{cat(app)}$ (s$^{-1}$) | $K_{m(app)}$ (mM) | $(k_{cat}/K_m)_{(app)}$ (mM$^{-1}$ s$^{-1}$) |
|---|---|---|---|---|---|
| Wild-type | IMP | 5 | 10.3 ± 1.9 | 0.34 ± 0.09 | 29.8 ± 9.6 |
| | | 8 | 13.5 ± 0.8 | 66.0 ± 6.3 | 0.21 ± 0.02 |
| | AMP | 5 | 23.2 ± 1.5 | 105.5 ± 13.4 | 0.22 ± 0.03 |
| | | 8 | 12.5 ± 0.9 | 72.7 ± 2.6[b] | 0.17 ± 0.01 |
| | pNPP | 8 | 0.0026 ± 0.0001 | 5.9 ± 0.5[b] | 0.00045 ± 0.000042 |
| | IMP (ATP)[c] | 8 | 21.0 ± 1.1 | 6.8 ± 1.1 | 3.1 ± 0.5 |
| | AMP (ATP)[c] | 8 | 19.4 ± 0.6 | 71.7 ± 2.8[b] | 0.27 ± 0.01 |
| | pNPP (ATP)[c] | 8 | 0.0037 ± 0.0005 | 5.3 ± 0.7[b] | 0.0007 ± 0.00013 |
| ΔN30 | IMP | 5 | 54.8 ± 5.7 | 2.8 ± 0.7 | 19.6 ± 5.3 |
| | | 8 | 28.3 ± 0.3 | 29.9 ± 1.8 | 0.95 ± 0.06 |
| ΔC10 | | 5 | 55.1 ± 2.5 | 1.6 ± 0.3 | 34.4 ± 6.6 |
| | | 8 | 35.5 ± 1.2 | 20.7 ± 0.6 | 1.71 ± 0.08 |
| K41L | | 5 | 22.5 ± 4.4 | 3.2 ± 1.3 | 7.0 ± 3.2 |
| | | 8 | 14.5 ± 0.8 | 15.6 ± 2.8 | 0.93 ± 0.17 |
| H398V | | 5 | 32.5 ± 3.4 | 1.6 ± 0.6 | 20.3 ± 7.9 |
| | | 8 | 16.5 ± 0.8 | 10.9 ± 1.0 | 1.5 ± 0.2 |
| D172N | pNPP | 8 | 0.28 ± 0.01 | 1.51 ± 0.15 | 0.19 ± 0.02 |
| D172A | | 8 | 0.019 ± 0.003 | 4.8 ± 1.4 | 0.0039 ± 0.0013 |

[a]Values represent mean and SD of the fitted parameters obtained from three independent measurements.
The plots from which the parameters in this table were derived are shown in Supplementary Figs. 6 and 7. The assays at pH 8.0 were carried out in 50 mM Tris-HCl, 30 mM MgCl$_2$, at pH 5.0 in 50 mM MES, 30 mM MgCl$_2$. The reaction mixture containing the substrate was mixed with 0.4 μM enzyme and incubated for 5 min at 25 °C. The liberated phosphate was estimated as described in "Methods". Reactions with pNPP as substrate were continuously monitored at 405 nm.
[b]The values are $K_{0.5}$.
[c]4 mM ATP.

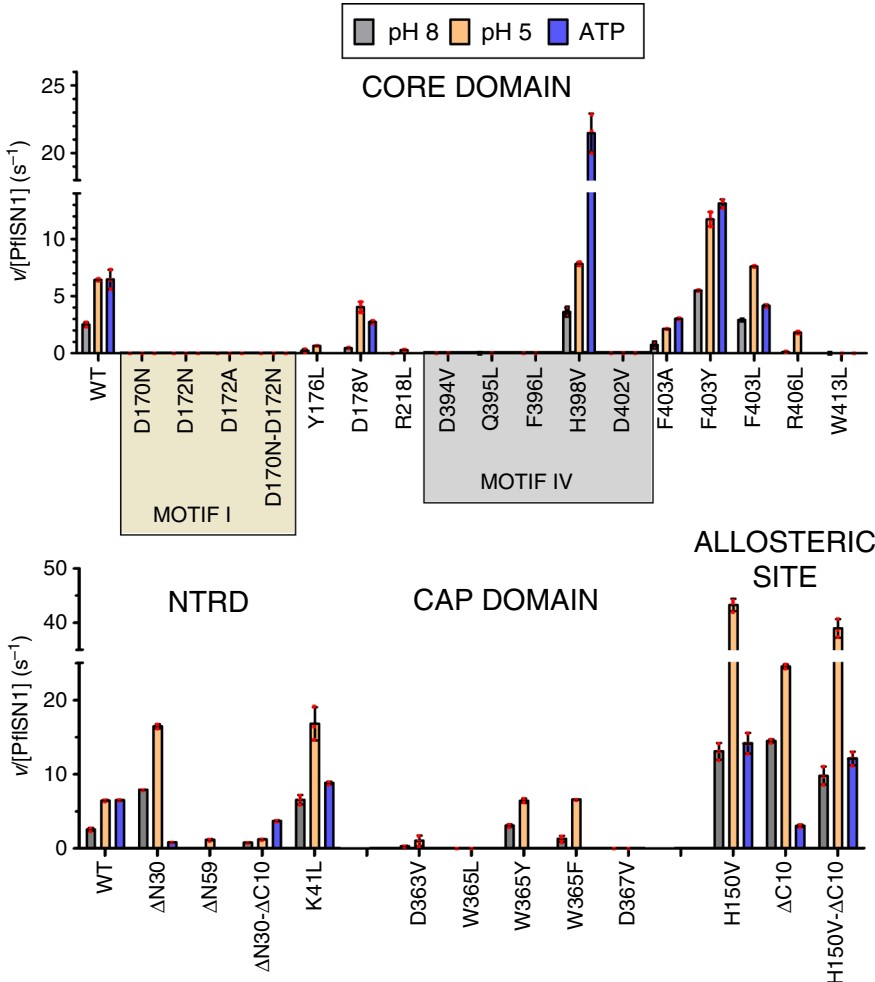

**Fig. 3 IMP-hydrolytic activity of *Pf*ISN1 and mutants at pH 8.0 and pH 5.0 and with ATP.** Histograms and error bars represent mean and SD, respectively, from three independent measurements. Location of mutations in *Pf*ISN1 is indicated above the histograms. *v*, the initial rate is the change in concentration of the product over time. NTRD, N-terminal regulatory domain. The concentration of enzyme and IMP used in all assays was 0.9 μM and 10 mM, respectively. Concentration of ATP was 2 mM. The conditions used for the assays are as mentioned in Supplementary Methods. WT, wild-type. Source data are provided as a Source data file.

and 2,3-BPG to an allosteric site[21–23]. Amongst the compounds screened at pH 8.0 (Supplementary Table 4), only ATP was found to be an activator (Supplementary Fig. 10A), a feature also observed for *Saccharomyces cerevisiae* ISN1[9]. With an affinity of $3.8 \pm 0.7$ mM, ATP (Supplementary Fig. 10B) is a K-type activator lowering the $K_{m(app)}$ value for IMP tenfold, while $k_{cat(app)}$ remains unchanged (Table 1, Supplementary Fig. 7K). ATP activation of IMP hydrolysis by *Pf*ISN1 was not observed at pH 5.0, or on the substrates AMP and pNPP (Supplementary Fig. 7K, M). In contrast to ATP activating *Pf*ISN1, PC and D-myo-inositol-4-phosphate showed 79% and 87% inhibition of enzyme activity, respectively (Supplementary Fig. 11).

**Pre-activation by ATP binding.** In the tetrameric structure of the ATP-bound wild-type enzyme (*Pf*ISN1-ATP), only two ATP molecules located in the two subunits having unstructured NTRDs were observed. The nucleotides bind to a cleft formed by helix α6 from the OD and helices α7 and α16 from the catalytic domain (Fig. 2b). ATP binds at the very place where the C-terminal loop (Loop$_{L430-Q444}$) is situated in the *Pf*ISN1-Apo structure (Fig. 4c), implying that this later must leave the cleft upon binding. A π-stacking occurs between H150 and the ade-nine moiety of ATP, and H150 establishes a bridge between E419

(catalytic domain) and Y129 of the OD (Fig. 4c). In presence of ATP, no electron density was observed for Loop$_{L430-Q444}$, sug-gesting it to be destabilized. Interestingly, in the two subunits showing a structured NTRD but lacking ATP, Loop$_{L430-Q444}$ stays in the cleft. Upon ATP binding, the enzyme undergoes a con-formational change where CD and Cap domains in both β and δ subunits bend, thereby adopting a more closed conformation than that seen for the wild type (Supplementary Movie 1, whole enzyme RMSD = 1.56 Å). Due to the interactions at interface 1, the ATP induced closure of subunits β and δ forces subunits α and γ to open slightly. Overall, the structures reveal that ATP activates the enzyme by triggering the departure of the Loop$_{L430-Q444}$ which in turn induces the bending of the subunit thereby increasing the affinity for IMP.

The *Pf*ISN1-ΔC10 mutant truncated by ten residues at the C-terminus showed eightfold higher catalytic efficiency ($k_{cat(app)}/K_{m(app)}$) at pH 8 when lacking ATP and no change at pH 5.0 (Table 1 and Fig. 3). In the presence of ATP at pH 8, the activity was highly reduced. The structure of *Pf*ISN1-ΔC10 displays a *Pf*ISN1-ATP-bound like conformation (RMSD = 0.36 Å) confirming the role of Loop$_{L430-Q444}$ as activity regulator. The ATP inhibiting effect on this variant suggests that the lack of Loop$_{L430-Q444}$ allows binding of four ATPs at a time instead of two, resulting in a non-physiological intermediate. Upon mutation of H150 to V,

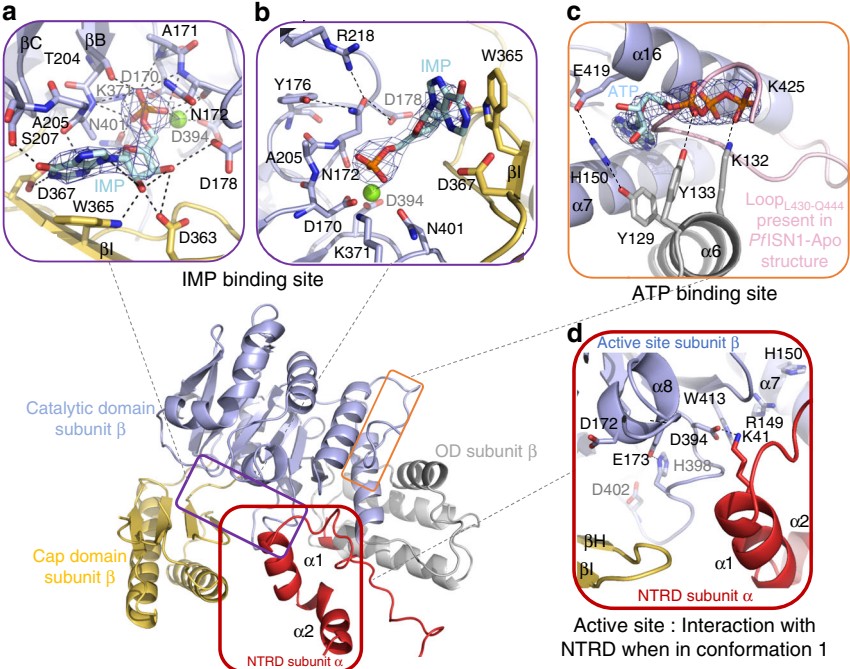

**Fig. 4 Structural features of *Pf*ISN1. a, b** Structural organization of the IMP binding site in the substrate binding state. **c** ATP-binding site in a pre-activated state with the C-terminal loop from the apo structure superposed. **d** Inter-domain communication of NTRD when this latter is in conformation 1.The 2*Fo*–*Fc* electron-density maps (blue mesh) are contoured at 1σ and $Mg^{2+}$ is depicted as a green sphere.

*Pf*ISN1$_{H150V}$-ΔC10 is no longer inhibited by ATP and *Pf*ISN1$_{H150V}$-Apo is not activated by ATP (Fig. 3). These observations confirm the role of H150 in ATP binding. Indeed in *Pf*ISN1$_{H150V}$-Apo and *Pf*ISN1$_{H150V}$-ΔC10, V150 can no longer interact with E419 and Y129, resulting in a reorganization of the effector site and an increase of the activity at pH 8.0 and 5.0 (Fig. 3 and Supplementary Fig. 12).

**Induced fit binding of IMP/$Mg^{2+}$.** In *Pf*ISN1-Apo, helix α8 (residues 171–175) carrying one of the catalytic aspartates, occupies the binding site of the phosphate moiety and $Mg^{2+}$. In this conformation, E173 (Motif 1) interacts with H398 (Motif IV) while D178 and D394 are exposed to the solvent. After IMP-$Mg^{2+}$ or $Mg^{2+}$ binding (*Pf*ISN1$_{D172N}$-IMP, *Pf*ISN1$_{WT}$-$Mg^{2+}$), helix α8 restructures into a loop and orients the E173 side chain to accommodate the substrate. Concurrently, side chains of D178 and D394 flip into the catalytic pocket to coordinate the substrate while the breakage of the salt bridge between E173 and H398 induces a complete reorganization of the segment between D394 and F407 (Fig. 5 and Supplementary Fig. 13) engendering α-helix D402-F407. Concomitantly, the four subunits bend even more than in the *Pf*ISN1-ATP conformation to adopt the most closed form observed (Supplementary Movie 1). In this latter conformation, we do not observe the NTRDs as seen in *Pf*ISN1-Apo and *Pf*ISN1-ATP structures. Indeed, the four NTRDs are relocated at interfaces 2 beneath Segment$_{D394-F407}$ (Fig. 5). As concerns Loop$_{L430-Q444}$, the four subunits display no electron density after residue L430 as seen for the ATP-bound subunits in the *Pf*ISN1-ATP structure. Despite adding ATP prior to crystallization, ATP molecules are not observed in *Pf*ISN1$_{D172N}$-IMP structure. The flip of F396 that occurs after the reorganization of Segment$_{D394-F407}$, allows a long-distance stacking with W413, R149 and H150. H150 slightly rotates, thereby breaking a hydrogen bond between H150 and Y129, and participating in remodeling the effector site as well as in the disruption of the π-stacking between H150 and the base moiety of ATP when

present. All these observations infer that the induced fit mechanism is triggered via a structural reorganization of Segment$_{D394-F407}$. This latter forms a long-distance interaction with effector site residues (Supplementary Fig. 12) inducing either the departure of Loop$_{L430-Q444}$ in absence of ATP or the release of ATP molecules if present, thereby allowing the closure of the enzyme and the correct accommodation of IMP by the Cap domain. As concerns the IMP-binding stoichiometry, we observed that in the *Pf*ISN1$_{D172N}$-IMP structure, each subunit binds one IMP-$Mg^{2+}$, a feature also observed in the *Pf*ISN1$_{D172N}$-ΔN30-IMP structure. IMP-$Mg^{2+}$ binding examined by isothermal titration calorimetry (ITC) on *Pf*ISN1$_{D172N}$ mutant showed a stoichiometry of 1:1, confirming the structural observations (Supplementary Fig. 14).

Analysis of the IMP hydrolyzing activity of Segment$_{D394-F407}$ mutants D394V, Q395L, F396L, H398V, D402V, F403A, F403L, F403Y and R406L (Table 1 and Fig. 3) emphasizes the importance of this structural reorganization. Among these, Motif IV residues D394, F396 and D402 are invariant across ISN1 sequences while Q395 and H398 are conserved. In the IMP-bound structure, Q395 coordinates $Mg^{2+}$, F396 flips upon IMP binding and forms a long-distance stacking interaction with H150, which in turn interacts with ATP, while H398 adopts alternate conformations in apo- and IMP-bound forms. Mutants Q395L and F396L were inactive on IMP at both pH 8.0 and 5.0 (Fig. 3). At pH 8, the $k_{cat(app)}$ of H398V does not change but a sixfold drop in $K_{m(app)}$ leads to a sevenfold increase in catalytic efficiency. At pH 5, the catalytic efficiency does not change as the threefold increase in $k_{cat(app)}$ is offset by a fivefold increase in $K_{m(app)}$ (Table 1). The activation of this mutant by ATP is threefold higher than that of the wild-type enzyme (Fig. 3). F403, adjacent to motif IV, displays different side-chain conformations across the structures of the *Pf*ISN1-Apo monomer with structured NTRD, with unstructured NTRD and *Pf*ISN1$_{D172N}$-IMP. The IMP hydrolyzing activity of F403A was compromised at both pH values and in the presence of ATP, while F403L had activity comparable to that of the wild type though activation by ATP was twofold lower. F403Y showed

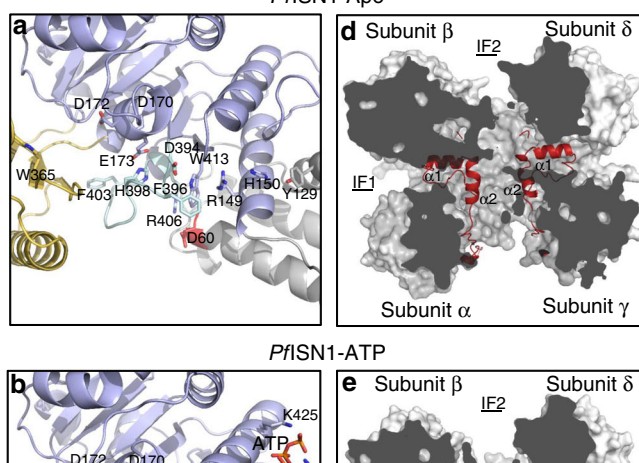

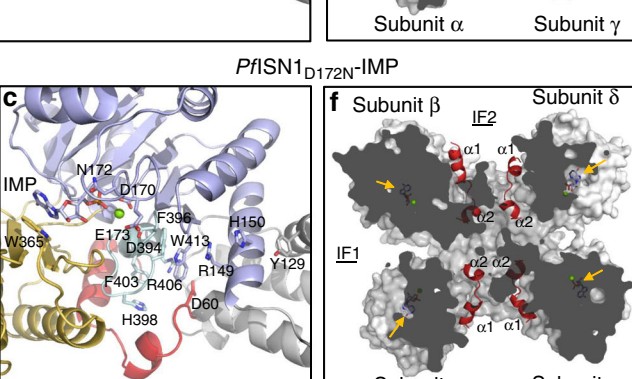

**Fig. 5 Reorientation of the NTRD. a**, **d** From apo- to **b**, **e** ATP-bound- and to **c**, **f** IMP-bound conformations. Blue arrows indicate the ATP-binding site, and yellow arrows the IMP-binding site. $Mg^{2+}$ is depicted as a green sphere in (**c**) and (**f**). IF1 interface 1. IF2 interface 2.

enhanced activity under all three conditions (Fig. 3). These observations suggest that F403 participates in the reorganization of the catalytic site and contributes to stabilize helix D402-F407 (Supplementary Fig. 15).

**Inter-domain communication by NTRD reorganization.** Three different conformations ("1", "2" and "3") are observed for the NTRDs, in which conformations 1 and 2 are both present in *Pf*ISN1-Apo and *Pf*ISN1-ATP, while conformation 3 is only seen in *Pf*ISN1-IMP (Fig. 5d–f, see Fig. 2 for interface definitions).

NTRDs in conformation 1 are entirely structured as observed in α and γ subunits of *Pf*ISN1-Apo and *Pf*ISN1-ATP (Fig. 4d, Fig. 5d, e, Supplementary Fig. 16), being stabilized by several hydrogen bonds and two salt bridges (K41-D394 and D60-R406). In conformation 2, NTRDs are not visible in the electron density from residues 1–59 as in the case for β- and δ subunits of *Pf*ISN1-Apo and *Pf*ISN1-ATP, but floating in the solvent (Fig. 5d, e) as suggested by SAXS data (Supplementary Fig. 5). In the *Pf*ISN1-Apo structure, K41 of NTRDs in conformation 1 interacts with D394 of the subunits displaying NTRDs in conformation 2. K41L shows a threefold activity increase at both pH 8.0 and pH 5.0, with no activation by ATP (Fig. 3). This mutant exhibits a

fourfold drop in $K_{m(app)}$ at pH 8, whereas at pH 5 the $K_{m(app)}$ value increases by tenfold (Table 1).

In conformation 3, all four NTRDs are visible for residues 36–59 and are located at interfaces 2 beneath $Segment_{D394-F407}$ (Fig. 5f). Helix α2 undergoes a ~135° rotation dragging concurrently the rest of the NTRD to slip at interface 2. Moreover, NTRDs have an impact on structuring $Segment_{D394-F407}$ (Fig. 5c, Supplementary Figs. 15, 16). Along with the NTRDs in conformation 1, $Segment_{D394-F407}$ forms a loop stabilized by D60-R406 and E173-H398 salt bridges. When the NTRDs are in conformation 2, salt bridge D60-R406 is broken, initiating the formation of α-helix D402-F407 (Supplementary Fig. 16). After $IMP-Mg^{2+}$ binding, $Segment_{D394-F407}$ is fully structured into α-helix D402-F407, which is stabilized by the NTRD α2 helix (Supplementary Fig. 15). Upon formation of helix D402-F407, salt bridge D60-R406 is reformed, probably participating in the structural reorganization of the enzyme after IMP hydrolysis. This is supported by a 24- and 4-fold activity drop of R406L at pH 8 and pH 5, respectively (Fig. 3). Moreover, deletion of the NTRDs (*Pf*ISN1-ΔN59) inactivates the enzyme at pH 8, while at pH 5 there is a sixfold drop in activity. Deletion of the 30 N-terminal residues (*Pf*ISN1-ΔN30) leads to a threefold increased activity at both pH 8 and pH 5 (Table 1 and Fig. 3). This deletion mutant at pH 8 shows twofold increase in $k_{cat(app)}$ with a similar fold lowering of $K_{m(app)}$ value. However, at pH 5, though the $k_{cat(app)}$ is increased by sixfold, the affinity for the substrate is compromised (eightfold enhancement in $K_{m(app)}$, Table 1). The structure of *Pf*ISN1$_{D172N}$-ΔN30 co-crystallized with IMP (*Pf*ISN1$_{D172N}$-ΔN30-IMP) is similar to the *Pf*ISN1$_{D172N}$-IMP structure (RMSD = 0.46 Å), confirming that only residues 30–59 are mandatory for stabilizing $Segment_{D394-F407}$ in conformation 3. Activity of *Pf*ISN1-ΔN30-ΔC10 under the three conditions is almost completely or significantly inhibited (Fig. 3), suggesting that the role of residues 1–29 is to stabilize the enzyme after ATP activation. Indeed, the structure of *Pf*ISN1-ΔN59 co-crystallized with IMP shows that when lacking NTRDs, $Segment_{D394-F407}$ cannot be stabilized (not well defined in the electron density), and consequently the enzyme is no longer able to accommodate the substrate correctly (IMP not observed in the crystal structure).

## Discussion

Amongst the three kingdoms of life, ISN1 is present only in eukaryotes with 83% of the selected sequences being confined to fungi. This apart, a small number of organisms belonging to the *Alveolata*, *Viridiplantae*, *Stramenopiles, Ichthyosporea* and *Cryptophyta* also possess ISN1. Interestingly, oomycetes belonging to the Stramenopile phylum form a sister clade with *Viridiplantae*, whereas diatoms and heterokont algae also belonging to Stramenopiles, form a different branch. *Vitrella brassicaformis*, a free living photosynthetic alveolata and an ancestor of *Plasmodium* also carries the gene for ISN1 (Supplementary Fig. 17). The cN-II class of purine nucleotidases is present in many eukaryotes and prokaryotes. Though the reaction performed is the same, ISN1s and cN-IIs have feeble sequence similarity (~10% identity between *Pf*ISN1 and human cN-II) as also reflected in the tertiary structures, which are not comparable. Indeed, comparative studies of *Pf*ISN1 with existing crystal structures revealed that neither the NTRDs nor the $Loop_{L430-Q444}$ are conserved among the known structures of the HAD phosphatase family. The OD, despite being a common feature of all ISN1s, is not conserved within the HAD phosphatase family, suggesting a role in quaternary structure organization being specific to the ISN1 sub-family.

Structure- and functional analyses revealed the essential role of NTRDs in the catalytic mechanism of *Pf*ISN1. We demonstrated

that when lacking NTRDs as in *Pf*ISN1-ΔN59 the activity is highly compromised. This is likely due to its incapacity to stabilize Segment$_{D394-F407}$ containing the ubiquitous HAD family motif IV, which is mandatory for coordinating IMP-Mg$^{2+}$. We also showed that whereas residues 30–59 are directly involved in the catalytic mechanism, residues 1–29 rather contribute to a global stabilization of the structure during conformational changes associated with catalytic function. Furthermore, the *Pf*ISN1-IMP structure shows a complete reorganization of the IMP-Mg$^{2+}$-binding site which also implies the structuring and stabilization of NTRDs. Hence, IMP-Mg$^{2+}$ binding induces both the positioning of the NTRDs within the tetramer and the closure of the four subunits.

Screening of potential substrates and allosteric effectors confirmed the substrate specificity of the enzyme for IMP with ATP as an activator and PC and D-myo-inositol-4-phosphate as inhibitors. ATP activation was observed only at physiological pH of 7–8 and not at lower pH values. Despite exhibiting maximum activity at acidic pH (4.0–5.0) and not at the physiological pH, in *P. falciparum* (asexual stages and gametocytes) ISN1 was found to be localized to the cytosol by both immunofluorescence and live-cell microscopy of episomally expressed ISN1-GFP fusion protein.

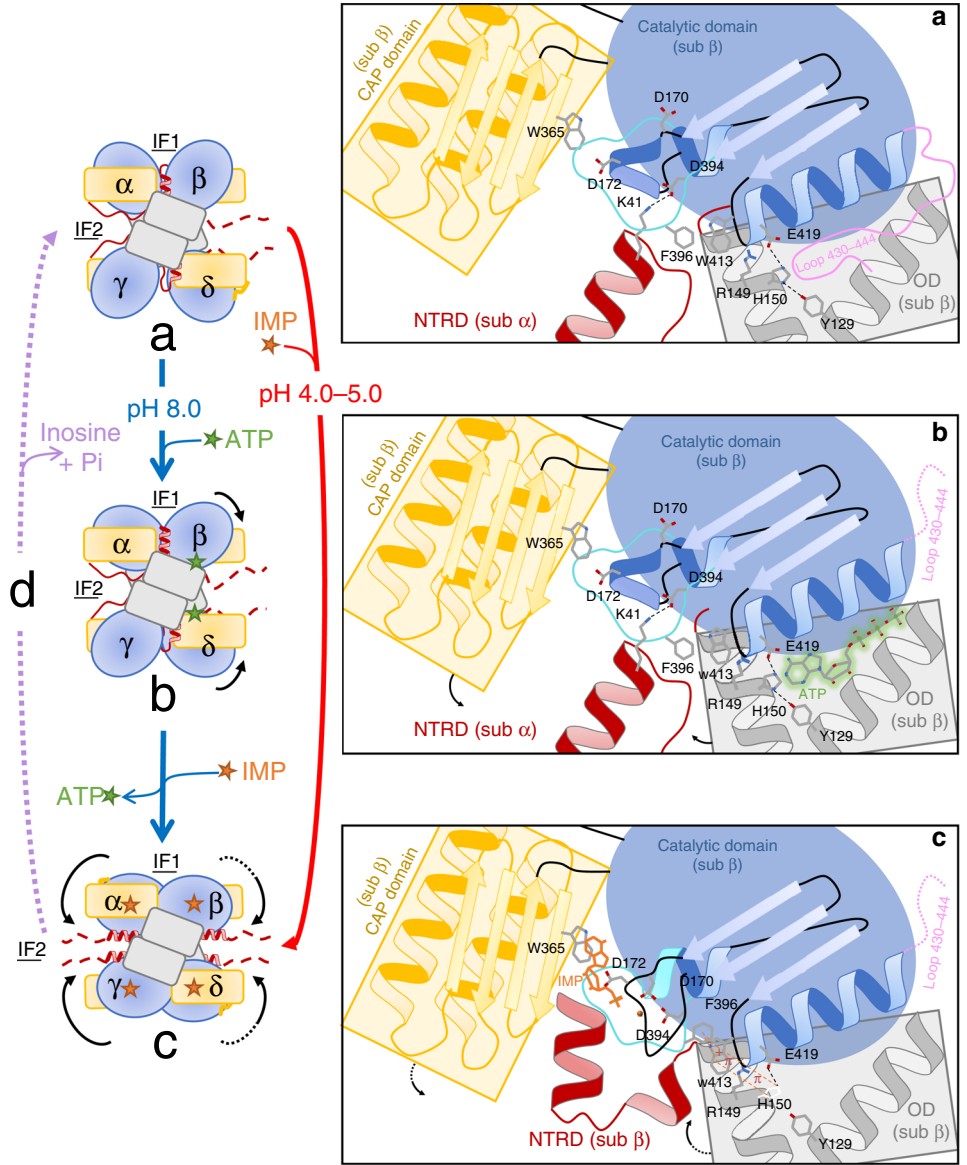

**Fig. 6 Model illustrating the catalytic mechanism of *Pf*ISN1.** In the first step 1 (**a** *Pf*ISN1-Apo), helix α8 carrying one of the catalytic aspartates (D172), occupies the binding site of the phosphate moiety and Mg$^{2+}$ and K41 from subunits α and γ NTRD form salt bridges with D394 from subunits β and δ, respectively. In a second step and at pH 8.0: ATP occupies only two of the four effector sites in the tetramer (subunits β and δ) (**b**) and induce a closure of these subunits upon ATP binding, thereby activating the enzyme after departure of Loop$_{L430-Q444}$. In this conformation, the base moiety of ATP forms a π-stacking with H150 which slightly turns and is stabilizes via hydrogen bond and salt bridge, respectively, with Y129 and E419. (**c**) Once activated and in a third step, the enzyme can bind IMP and Mg$^{2+}$. This binding induces both reorganization of helix α8 and Segment$_{D394-F407}$. As a consequence, F396 flips engendering long-distance stacking with W413, R149 and H150. This latter slightly rotates, breaks the hydrogen bond with Y129, causing departure of ATP. Closure of the enzyme induces correct accommodation of IMP by the Cap domain and forces the four NTRDs to relocate at interface 2. At pH 4.0–5.0: ATP is not an allosteric activator. IMP can bind to the four subunits allowing the enzyme to go directly from step 1 (**a**) to step 3 (**c**). Common for all pH values: step 4 (**d**) Return to the initial apo conformation including Loop$_{L430-Q444}$ returning in the effector binding sites and release of inosine and $P_i$ into the medium. sub subunit, IF interface; black arrows indicate global movement of domains. colour code used is as in other figures.

This localization was also seen in *P. berghei* when *Pf*ISN1-GFP was episomally expressed. Cytosolic 5′-nucleotidases from other organisms have been reported to have pH optimum in the range of 5.0–7.0[24,25]. The food vacuole, in which haemoglobin degradation takes place, constitutes the major acidic compartment in *Plasmodium*. The multiple strategies used for localization in this study do not show localization in this acidic compartment.

Overall, our results suggest a reaction mechanism (Fig. 6) in which, at physiological pH (Step 1), ATP occupies only two of the four effector sites in the tetramer (Step 2), suggesting an asymmetric binding nature in which IMP-$Mg^{2+}$ preferentially bind to the two pre-activated subunits. In addition, the two ATP-containing subunits have their NTRDs in conformation 2, ready to re-structure into conformation 3 to stabilize Segment$_{D394-F407}$ when binding IMP-$Mg^{2+}$. The appearance of IMP-$Mg^{2+}$ in the active site triggers the reorganization of Segment$_{D394-F407}$, during which F396 flips to form a long-distance stacking with W413, R149 and H150 reorienting the imidazole ring of this histidine (Step 3). This structure based conclusion is supported by the fact that *Pf*ISN1$_{F396L}$ and *Pf*ISN1$_{W413L}$ are inactive while H150V is not activated by ATP. Consequently, the hydrogen bond between H150 and Y129 as well as the "π-stacking" formed between H150 and the nitrogen base of ATP breaks to induce ATP release into the medium. The destabilization of interactions at the effector site induces the complete closure of the four subunits allowing the Cap domain to stack the nitrogen base of IMP with W365 (Fig. 6, Supplementary Figs. 12 and 18), supported by the lack of activity of W365L. Simultaneously, binding of IMP-$Mg^{2+}$ results in a breakage of the salt bridge D394-K41 of the other subunit (Supplementary Fig. 16). The closure of the Cap domains engenders a steric clash with the two NTRDs in conformation 1, forcing these latter to reorient into conformation 2 (Fig. 5, Supplementary Fig. 16). At the same time, hydrolysis of IMP-$Mg^{2+}$ requires the reopening of the enzyme in order to release the newly formed inosine. In this conformation, a salt bridge between D60 and R406 is formed, while the formation of the phosphoaspartyl-enzyme intermediate allows Segment$_{D394-F407}$ to initiate reorganization, in particular the flipping of F403. This latter most likely participates in the reopening of the Cap domain and/or to the return of the NTRDs in conformation 1 due to steric clashes with W47. The importance of salt bridge D60-R406 in the catalytic cycle is corroborated by mutant R406L for which the activity is highly compromised. A return to the *Pf*ISN1-Apo conformation may allow the release of inosine to the medium, and a sub-sequent hydrolysis of the phosphoaspartyl-enzyme intermediate. This conformation is stabilized by the return of loop$_{L430-Q444}$ into the effector sites (Step 4).

While ISN1 is present in *P. falciparum* and certain other species of primate and avian malarial parasites, *P. berghei* lacks a homologue of this enzyme. This could be attributed to metabolic differences between *Plasmodium* species or differences in host–parasite interactions, restricting it to parasites, which only infect a certain class of hosts[26]. *Pf*ISN1 was also reported to be one amongst a set of host-specific lysophosphatidylcholine (LysoPC)—responsive genes that were transcriptionally upregulated in response to depletion of host LysoPC levels, initiating sexual commitment and gametocytogenesis in *P. falciparum*[11]. Although the ISN1 gene is mutable during the erythrocytic stages, the significance of its upregulation in female gametocytes needs further studies.

## Methods

### Phylogenetic and molecular evolutionary analyses
Phylogenetic and molecular evolutionary analyses of ISN1 sequences were conducted using MEGA v7[27].

### *Plasmodium* transfection and microscopy
The in vitro culture of the erythrocytic stages of *P. falciparum* was maintained as described by Trager and Jensen[28]. Gametocyte production and enrichment was done as described by Fivelman et al.[29] and details are provided in Supplementary Methods. *P. berghei* ANKA parasites were maintained in BALB/c mice. *Pf*ISN1 gene was obtained by reverse transcription-PCR on parasite RNA.

The sequences of primers used for PCR amplification are provided in Supplementary Table 5. *Pf*ISN1 gene was cloned into pFCENv1 and pBCEN5 to obtain a C-terminal fusion with GFP. The resulting plasmids pFCENv1_*Pf*ISN1 and pBCEN5_ISN1GFP were used for transfection of *P. falciparum* and *P. berghei*, respectively using established protocols[30,31] with details provided in Supplementary Methods. The confirmed lines of parasites carrying the *Pf*ISN1-GFP gene were examined by live-cell fluorescence microscopy using a Zeiss® LSM-510 META confocal microscope. Anti-*Pf*ISN1 antibody was raised in rabbits using purified recombinant *Pf*ISN1 and affinity purified using Sepharose beads conjugated to *Pf*ISN1. Using the antibody, indirect immunofluorescence microscopy was carried out using a Zeiss® LSM-510 META confocal microscope to check the localization of *Pf*ISN1 in intraerythrocytic stages of *P. falciparum*.

### Animal experimentation
All animal (male rabbit for anti-PfISN1 antibody generation and BALB/c strain of mice for maintenance of *P. berghei* ANKA WT and PfISN1-GFP expressing parasites) experiments adhered to the standard operating procedures prescribed by the Committee for the Purpose of Control and Supervision of Experiments on Animals (CPCSEA) and approved by the Institutional animal ethics committee (IAEC) of the Jawaharlal Nehru Center for Advanced Scientific Research. IAEC comes under the purview of CPCSEA.

### Expression and purification of *Pf*ISN1 and mutants
Cloning of wild-type and mutant *Pf*ISN1 into pET-21bN was carried out using standard procedures as described in Supplementary Methods. For protein expression, either BL21 (DE3) or Rosetta strains were used. The cell pellet was re-suspended in 30 ml lysis buffer and lysed using French© pressure cell press (Thermo IEC Inc., USA) over six cycles at 1000 psi. The components of the lysis buffer were 50 mM Tris-HCl, pH 8.0, 100 mM NaCl, 10% (w/v) glycerol, 0.1 mM PMSF and 0.5 mM Tris-(2-carboxyethyl) phosphine. The lysate was centrifuged at $14,000 \times g$ for 45 min at 5 °C and the supernatant bound to Ni-NTA agarose beads (NI-NTA His-Bind® Resin, Qiagen) for 3 h at 5 °C. Post binding, the beads were loaded onto a glass column and washed with at least ten equivalent of bead volume of lysis buffer containing increasing concentrations of 0, 20 and 40 mM imidazole. The protein was eluted in 5 mL of lysis buffer containing 500 mM imidazole. One millimolar of EDTA was added to chelate $Ni^{2+}$ ions that could have eluted along with the protein. The eluted protein was concentrated using Amicon® Ultra Centrifugal filter with a 30 kDa molecular weight cut-off (Millipore™ Corporation) and loaded onto a 16 mm × 60 cm column packed with Sephacryl™ S-200 HR (GE Healthcare Life Sciences).

### Enzyme kinetics and characterization
Enzyme activity assays were carried out using Chen's method[32] to estimate the liberated phosphate. All assays on phosphorylated metabolites were carried out at pH 8.0 and 5.0. Continuous spectrophotometric assay involving the monitoring of change in absorbance at 405 nm was used to evaluate the hydrolysis of pNPP. Phosphotransferase activity of *Pf*ISN1 using 5′-IMP-adenosine as donor–acceptor pair was monitored by ion-paired reverse-phase HPLC.

Stoichiometry of ligand binding was determined for the complex of *Pf*ISN1$_{D172N}$ mutant and the ligand IMP using a VP-ITC. Details of assay conditions are described in Supplementary Methods. Generation of all plots and data fitting by non-linear regression were done using GraphPad Prism® version 5.0 (GraphPad Software Inc., San Diego, CA). The equations used for fitting the data and selection of the model that fitted best are elaborated in Supplementary Methods. Details of all procedures used are provided in Supplementary Methods.

### Protein production for structural studies
Protein production and purification for structural studies was carried out as described in detail in Supplementary Methods. The plasmid carrying the different constructs of *Pf*ISN1 gene was transformed into *Escherichia coli* BL21-CodonPlus (DE3)-RIL cells, except for the selenomethionine-derivative for which the plasmid carrying the *Pf*ISN1 gene was transformed into *E. coli* B834 (DE3) strain.

### Crystallization and structure determination
Crystallization conditions screening was carried out at 292 K (vapour-diffusion in sitting-drops), using commercially available crystallization kits. For screening, a Mosquito® crystallization robot from TTP Labtech was employed using two protein/crystallization agent ratios (200 + 200 nL and 300 + 100 nL drops equilibrated against 70 μL in MRC Crystallization Plates (Molecular Dimensions)). Proteins were concentrated to 13 mg mL$^{-1}$ in 50 mM Tris-HCl pH 8.0, 100 mM NaCl buffer except for the screen resulting in crystals at acidic pH and for which the protein was stored in 50 mM MES pH 5.0 and 100 mM NaCl.

X-ray diffraction data on a selenomethionine-derivative crystal (crystallized in the presence of ATP) was collected (ID29 - ESRF, Grenoble, France) at a

wavelength of 0.979230 Å for SAD phasing. Phases and experimental electron density maps were calculated with the Phenix AutoSolprogramme[33] and the initial model was built using Phenix AutoBuild employing the *Pf*ISN1-ATP data for phase extension. All the remaining structures of *Pf*ISN1 were solved by molecular replacement using the ATP-bound structure as search model.

**Small-angle X-ray scattering**. For collection of SAXS data, samples were concentrated to ~13 mg mL$^{-1}$. Initial processing was done using DataSW[34] and PRIMUS and P(r) analysis was carried out using GNOM. Ab initio models of *Pf*ISN1 were generated from the experimental data using both DAMMIN and GASBOR. The 50 generated models were averaged and filtered using the DAMAVER program suite to generate the final model[35] and the AllosMod-FoXS server[36].

**Negative staining electron microscopy and image analysis**. *Pf*ISN1 samples at 0.025 mg mL$^{-1}$ were applied to formvar grids and stained with 2% (w/v) sodium silicotungstate, pH 7.0. Images were recorded with a Tecnai spirit microscope operating at 120 kV, at nominal ×49,000 magnification with a pixel size of 1.36 Å. Approximately 19,000 individual particles of *Pf*ISN1 were automatically picked using the LoG picker in Relion 3.0 before 2D classification. Good looking classes were selected and submitted to a round of 3D classification. The best-looking map containing 2600 particles was finally refined to 20 Å resolution imposing D1 symmetry. To predict the functional motions of *Pf*ISN1, NMA calculations were performed with the ProDy package[37].

**Comparative studies of three-dimensional structures**. Crystal structures of *Pf*ISN1 were compared with existing structures in the protein data bank at Rutgers, RCSB, using the DALI server[38].

**Figure rendering**. Figures of three-dimensional structures were drawn with PyMol (DeLano Scientific LLC, http://pymol.sourceforge.net/) and Chimera[39].

**Reporting summary**. Further information on research design is available in the Nature Research Reporting Summary linked to this article.

## Data availability

Coordinates and structure factors have been deposited in the Protein Data Bank under accession codes 6RMO (*Pf*ISN1-Apo), 6RMD (*Pf*ISN1-ATP), 6RME (*Pf*ISN1$_{D172N}$-IMP), 6RNH (*Pf*ISN1-ΔC10), 6RMW (*Pf*ISN1$_{D172N}$-ΔN30-IMP) and 6RN1 (*Pf*ISN1-ΔN59), respectively. The source data underlying Figs. 1a–c, 3, and Supplementary Figs. 2A–C, 6A–F, 7A–M, 8A, B, 9A, B and 10A, B are provided as a Source Data file. Other data are available from the corresponding authors upon reasonable request. Source data are provided with this paper.

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

## Acknowledgements

Technical support from staff on beamlines MX and FIP (both European Synchrotron Radiation Facility), as well as on beamlines PX3 (Swiss Light Source, Switzerland) and Proxima 1 (Soleil, France) is gratefully acknowledged. We are also grateful for assistance from V. Gueguen-Chaignon and F. Delolme of the Protein Science Facility of SFR Biosciences Lyon (UMS3444/US8) and G. Schoehn from the Grenoble Instruct centre (ISBG; UMS 3518 CNRS-CEA-UJF-EMBL). The following reagents were obtained through BEI Resources Repository, NIAID, NIH: *Plasmodium falciparum*, Strain 3D7, MRA-102, contributed by D.J. Carucci; *Plasmodium falciparum*, Strain 3D7A, MRA-151, contributed by D. Walliker and *Plasmodium berghei*, Strain ANKA, MRA-671, contributed by M.F. Wiser. We acknowledge S. Iwanaga, Mie University, Tsu, Japan for providing us with the plasmids pbCEN5 and pFCENv1. This research was funded by the CNRS, the French Infrastructure for Integrated Structural Biology (FRISBI) ANR-10-INSB-05, the ANR project grant "PLASMOPUR" ANR-17-CE11-0032, the Région Rhone-Alpes (ARC1 Santé – ADR2013), the Fondation pour la Recherche Médicale (FRM: FDT20160435284) and the Fondation Innovation en Infectiologie, "FINOVI project AO12-27"). For HB this research was funded by the Department of Biotechnology, Ministry of Science and Technology, Government of India Grants BT/PR11294/BRB/10/1291/2014, BT/PR13760/COE/34/42/2015, and BT/INF/22/SP27679/2018; Science and Engineering Research Board (SERB), Department of Science and Technology, Government of India Grant EMR/2014/001276; JC Bose Fellowship, SERB and institutional funding from the Jawaharlal Nehru Centre of Advanced Scientific Research, Department of Science and Technology, India. A.S. acknowledges CSIR for Junior and Senior Research Fellowships, RR acknowledges CSIR for Junior Research Fellowship. The authors wish to thank the reviewers for their constructive and precious comments and suggestions.

## Author contributions

Conceptualization H.B., N.A; formal analysis, H.B., A.S., N.V., R.R., L.C., L.B., S.V., N.A.; funding acquisition H.B., N.A.; investigation L.C., L.B., S.V., A.S., N.V., R.R., B.S., U.T.G., S.K.; supervision H.B., L.B., N.A.; validation A.S., N.V., R.R., L.C., L.B., S.V.; writing H.B., A.S., N.V., R.R., L.C., L.B., S.V., N.A.; R.R. and N.V. contributed equally to this work.

## Competing interests

The authors declare no competing interests.
