## [Peer Review File · Nature Communications]

Reviewers' comments:

Reviewer #1 (Remarks to the Author):

This manuscript focuses on the ISN1 nuclease of Plasmodium parasites, understanding its cellular location, its structure and mechanism and its role nucleotide metabolism.

Figure 1 presents some well conducted localisation experiments in which both antibody mediated detection and GFP-labelling were used to show that ISN1 is expressed in the cytosol of different developmental stages of the parasite.

The authors then show that overexpression of ISN1-GFP leads to reduced parasitaemia in culture growth (Figure 1D). The authors speculate that this is because increased levels of ISN1 retard growth and suggest a possible role in gametocyte formation. This is a huge over-interpretation of what is a preliminary observation. Firstly, no controls were done to show that ISN1 overexpression is the cause of the reduced growth rates. Perhaps fusing GFP to ISN1 affects its function and thereby reduces growth? This type of dominant effect is possible for a tetrameric enzyme. Perhaps the transfection procedure has affected growth? If the authors want to make this conclusion, then they firstly need to do appropriate controls, such as to over-express untagged ISN1 and see if expression levels alter. Also, they needed to produce mock transfected cells and assess expression levels. Finally, they need to set up over-expressing in a strain that produces gametocytes and to measure the number of gametocytes or conditionally knock out ISN1 and see if it leads to loss of gametocytes.

The authors then determine the structures for a variety of constructs. The structures mostly look well determined for their resolution. But it is hard to be sure about the quality of the crystallographic data, or whether the resolution is correct as the R factors and CC1/2 appear to be missing from the crystallographic tables? These are essential parameters, without which the quality of the data cannot be judged. The models mostly appear good, although there are some refinement issues. Both 6MN1 and 6RMO have very high clash scores and Ramachandran outliers and these should be fixed. Otherwise the structures appear good and are well supported by SAXS and SEC-MALLS analysis. There is also a negative stain reconstruction, which fits the crystal structure. However, a SI figure is needed for this negative stain reconstruction, showing raw data, class averages and resolution.

The next section of the manuscript appears high quality and very informative, with nice figures. There is a strong set of catalytic data, identifying the substrates for this enzyme, and assessing its mechanism. There is also a very nice and detailed description of the different structural states of the enzyme in its complex with different substrates and a comparison of this with the mechanistic data. The discussion of this data, the many data tables and files and the well described methods are all praiseworthy and informative.

In summary, the strengths of this manuscript are the structural and catalytic data on an interesting enzyme expressed in human-infective Plasmodium species. This is all very well done and of high quality. The weakness of the manuscript is the claim of an important role for the over-expression of this enzyme in gametocyte development without any good evidence being presented. In my view the claims in the abstract (lines 31-35), results (lines 90-98) and conclusion (lines 366-389) are unsupported with data and should be removed. If the authors want to include these claims, they need to do the controls for Figure 1D, show the effect on gametocyte numbers of over-expression of untagged ISN1 and/or knock out ISN1 and see if it reduces gametocyte numbers. However, without these claims, this is a well-done experiment on the structure and mechanism of an interesting enzyme.

Minor points:

Line 76 What if ISN1 is present other Plasmodium species but just more divergent and hard to detect? Seems odd that it is in primate and bird infective species, but not rodent infective species? Why? Perhaps re-phrase.

Line 100-101 What is the rationale for the truncations?

Reviewer #2 (Remarks to the Author):

Carrique L et al. present a detailed characterization of the unusual inosine monophosphate nucleotidase PfISN1 from the human malaria parasite *Plasmodium falciparum*. The manuscript content comprises 7 crystal structures, kinetic data for a variety of substrates and mutants as well as localization and overexpression studies for PfISN1-GFP in *P. falciparum* and *P. berghei*. While the structure-function analyses are quite convincing, the presentation of the kinetic data leaves room for improvement. The *in vivo* significance of the gene/protein has not been addressed properly and the drawn conclusions/hypotheses are too speculative and should be toned down or even deleted. In my opinion, the manuscript rather suffers than benefits from the *in vivo* data because the emphasis of the physiological relevance is oversold by the authors. If the authors do not want to tone this aspect down and really want to spin the story the way they do - with an emphasis on the *in vivo* significance of PfISN1 in the title and throughout the manuscript - they have to perform knock-out or CRISPR/Cas9 mutagenesis studies. This would be, in my opinion, beyond the scope of the study. I therefore rather recommend to present the highly interesting structure-function analysis as is, improve the presentation/analysis/interpretation of the kinetic data and add the localization and potential phenotype for overexpressed PfISN1-GFP as a bonus without further ado.

Major points:

1) Phenotype

- page 5, lines 90 - 98 and other sections: The phenotype for the PfISN1 overexpressing 3D7 strain in Fig. 1D is questionable because of a) the study design, b) the lack of a significant phenotype for 4 out of 6 data points and c) the data presentation:

a) How can the authors exclude that the growth defect did not originate from the drug treatment (WR99210) or other plasmid-derived effects that were absent in 3D7? The growth should be compared with a cell line containing the empty or a GFP(only)-encoding plasmid as a control. Furthermore, overexpression of untagged PfISN1 should be also included as a control in order to interpret the phenotype and to exclude alternative effects. At the current stage the growth defect might have many possible reasons. One of several alternative explanations that the authors neglect is a potential inactivation of PfISN1 because of a dominant-negative of the GFP tag: The authors nicely showed that the C-terminus plays a role for the activity of tetrameric PfISN1. Hence, overexpressed PfISN1-GFP might actually interact with endogenous wild-type PfISN1 resulting in inactive heterooligomers and a decreased overall nucleotidase activity (instead of an expected increased activity in the overexpressing strain). The interpretation of the phenotype without controls with untagged (or N-terminally GFP-tagged) overexpressed PfISN1 is too speculative and the relevance of the nucleotidase for gametocyte development/drug development etc. should be toned down or even deleted.

b) How do the authors explain that the growth was apparently highly similar for 4 out of 6 data points? Maybe the difference became only relevant at a high parasitaemia that was coupled to an extreme consumption of nutrients (hypoxanthine) at day 6?

c) Potential growth differences at day 1-4 cannot be seen in the graph because of the way the data is presented. Furthermore, the choice of y-axis in Fig. 1D is confusing and unusual. The experiment was started with 1% parasitaemia at day 0 and resulted in a roughly 100-fold increased parasitaemia at day 6. Taking into account a 5-fold dilution at day 4, does 100% correspond to a (potentially crashed) culture with a parasitaemia of 20%? Please plot the y-axis as % parasitaemia without normalization so that the values for days 1-4 are visible and clearly indicate the 5-fold dilution for both cultures at day 4.

- PfISN1 was one out of 300 (!) genes that were upregulated in gametocytes in the study by Brancucci et al. A slight growth reduction of asexual blood-stage parasites that overexpressed PfISN1-GFP was observed in cell culture experiments without adequate controls. These two facts combined are not sufficient for a proof-of-principle or the generalized conclusions regarding the physiological significance of PfISN1 for gametocyte development. If the authors really want to make this extremely speculative connection (as emphasized in the summary and throughout the manuscript) they should a) determine the concentrations of IMP/AMP/GMP for strains with an empty plasmid as well as overexpressed PfISN1-GFP vs. overexpressed untagged PfISN1, b) quantify PfISN1 in these strains by western blotting using their antibody and various concentrations of recombinant PfISN1 for calibration, and c) quantify whether *P. falciparum* and *P. berghei* parasites that overexpress PfISN1 form more/less female/male gametocytes. If the authors are right, the purine nucleotide concentrations should be drastically altered depending on the PfISN1 concentration (as hypothesized by the authors, page 5, lines 94-98) and *P. falciparum* and *P. berghei* with overexpressed PfISN1 should form more/less female/male gametocytes.

- page 4, lines 76-81 and many other sections: If ISN1 is such a central metabolic enzyme, why was it presumably lost in all rodent malaria parasites? Furthermore, a check of the piggyBac insertion mutagenesis data in PlasmoDB suggests that PfISN1 is dispensable and that its loss has no fitness cost. Please cite the study (Zhang et al 2018 Science 360:6388) and take it into account regarding the central hypothesis on the relevance of PfISN1.

2) Kinetics

- Whole manuscript: The terms k_{cat} , K_m , k_{cat}/k_m , K_d etc. are incorrect because all values reflect apparent (and not true) kinetic parameters that were derived from primary plots at one constant parameter (e.g. one constant ATP and/or Mg^{2+} concentration). If the authors want to make claims about the true kinetic constants of the enzymes, they have to extrapolate the constants (from secondary plots) based on systematic variations of all relevant parameters. This would be quite some work. Thus, I rather recommend to replace all terms with apparent k_{cat} , apparent K_m etc. and to be much more careful about the interpretation of the constants (e.g., regarding substrate affinities under physiological conditions, the assignment of pH activity profiles and a macroscopic pK_a value to different ionisable groups in the substrate and enzyme etc.).

- Extended Data Fig. 4 and page 7: This part of the manuscript lacks clarity and is difficult to follow because of the incorrect figure legend. Poor data fits in panels B (pK_a extrapolation), panel C ([IMP] variation) and panel E ($[Mg^{2+}]$ variation) raise doubts about the used equations, kinetic models and interpretations for all indicated parameters (sumarized in Table 1). The kinetic data of the tetrameric enzyme are obviously too complex to be fitted using a simple M.-M. or Hill equation. Alternative equations, models etc. that might result in better fits can be found for the pK_a values in Brocklehurst K 1996 (Physical factors affecting enzyme activity. In: Enzymology Labfax 175-198) and for oligomeric proteins with modulators in Segel IH 1993 (Enzyme Kinetics: Behavior and Analysis of Rapid Equilibrium and Steady State Enzyme Systems). Please repeat the analyses with appropriate equations that are based on plausible kinetic models and correct the according paragraphs.

- What are the expected physiological concentrations of ATP and IMP in *P. falciparum* and how are

these concentrations reflected by the chosen assay conditions and sigmoidal curves in primary plots?

- Table 1: The primary plots and data fits for most of the data are not shown in the manuscript and should be included.

- Table 1 and all figures showing kinetic data: The exact conditions should be indicated (e.g., modulator concentration tested, enzyme concentration, temperature, salt concentrations, buffer concentrations etc.) for each set of experiments because the apparent constants can highly depend on them.

- Extended Data Fig. 6: Please indicate which type of inhibition the plot and fit are based on, why this type of inhibition was chosen and why the inhibition constant is interpreted as an apparent K_d value.

- Page 9 and Extended Data Fig. 7: The data in Fig. 7 does not show that ATP is a competitive (K -type) inhibitor. Please show the according M.-M. and Lineweaver-Burk plots for different IMP and AMP concentrations to support the statements on the apparent K_m and k_{cat} (V_{max}) values. Please indicate which equation/model/fit was used for the plot in Fig. 7B and how the "affinity" was determined.

- Page 10, line 203-205; Fig. 3 shows an activity plot. No data on the apparent catalytic efficiencies are shown.

3) Others

- The PDB reports for structures 6RMO, 6RN1 and 6RMW show numerous clashes and Ramachandran outliers. Please clarify.

- Extended Data Fig. 11B: Poor data fit. Does the fit take into account potential multiple binding sites/allosteric/cooperative effects for each subunit of the homodimer? There is no insert in contrast to the figure legend. Please clarify.

Minor points:

- page 2: Some parts of the summary are too vague or imprecise and do not really describe the results. The first sentence has nothing to do with the study and could be deleted. The statement that the structures "reveal complex domain organizations tightly associated with catalysis" could be more specific. The in vivo overexpression data is limited to PfINS1-GFP (not PfINSP) and the hypothesis on the metabolic switching at the end of the summary is too speculative considering the available data and the absence of ISN1 in numerous Plasmodium species.

- page 2, line 27: Please introduce the abbreviation ISN1.

- page 3, line 55: Maybe replace "homology" with "similarity" (two genes/proteins are, as far as I understand the concept, either homologous or not, but genes/proteins can have different degrees of similarity).

- References are missing to support the statements. For example:

page 3, line 42: "Purine nucleotides are mandatory... phosphotransferase activity." (4 sentences without reference)

page 3, line 56: "Plasmodia lack... including humans."

- page 5, line 86: Please indicate that the antibody was generated in this study against recombinant PfISN1.
- Please include quality controls for the new antibody (e.g., a western blot as extended Fig. 1D and a negative control in Fig. 1A).
- page 25, line 571: Please replace "episomal expression of ISN1" with "episomal expression of PfISN1-GFP" and indicate how the growth assay was started (synchronous ring-stage parasites at an initial parasitaemia of 1%?)
- Please explain to the unexperienced reader whether the PbEf1alpha promoter is a strong/weak constitutive/stage-dependent promoter in *P. falciparum*.
- The upper row in Fig 1B shows a trophozoite (not a ring stage) parasite.
- Page 5, line 104: Is there a reference for the domain nomenclature? If yes, please cite the reference(s).
- Extended Data Fig. 2: Please include data for a calibration curve, add the calculated molecular mass for monomeric recombinant PfISN1 to the figure legend and indicate the apparent molecular mass for the peaks at roughly 10.2 and 11.9 mL.
- page 7, line 129 and other sections, y-axis labels in Extended Data Fig. 4+8 etc.: Please do not use $\mu\text{mol substrate per min \& mg enzyme}$ (" $\mu\text{mol min}^{-1} \text{mg}^{-1}$ ") as units. The y-axis label " v " is incorrect. Please use the correct axis label " $v/[\text{enzyme xy}]$ ". The correct unit is " sec^{-1} ".
- page 8, line 150-152: Reference supporting statement missing.
- page 8, line 150-152: "on the synthetic compound pNPP, D172N and D172A mutants showed 150- and 25-fold increased activity". Where is this data shown?
- page 9 and Extended Data Fig. 8: Please indicate the conditions that lead to 79/87% inhibition.
- page 59, line 932: Do the duplicates represent independent expression and purification experiments (biological replicates) ?
- Extended Data Fig. 14: Aevolata (typo)
- Fig. 5: The panels in the upper row are too small. Maybe use a 3x2 instead of a 2x3 format?

Responses to reviewers (please notice that due to the changes that have been made, the original numbering given in questions from reviewers is no longer exactly the same)

General remark to reviewers: Please notice that due to the part that was requested to be deleted from the manuscript we have changed the title from “*Plasmodium falciparum* nucleotidase: catalytic regulation and in vivo significance” to “Structure and catalytic regulation of *Plasmodium falciparum* IMP-specific nucleotidase”

Reviewer #1 (Remarks to the Author):

This manuscript focuses on the ISN1 nuclease of Plasmodium parasites, understanding its cellular location, its structure and mechanism and its role nucleotide metabolism.

Figure 1 presents some well conducted localisation experiments in which both antibody mediated detection and GFP-labelling were used to show that ISN1 is expressed in the cytosol of different developmental stages of the parasite.

The authors then show than overexpression of ISN1-GFP leads to reduced parasitaemia in culture growth (Figure 1D). The authors speculate that this is because increased levels of ISN1 retard growth and suggest a possible role in gametocyte formation. This is a huge over-interpretation of what is a preliminary observation. Firstly, no controls were done to show that ISN1 overexpression is the cause of the reduced growth rates.

Although we do not have a direct measure of the levels of ISN1 in our parasite line carrying the plasmid, we do know that endogenous expression along with additional expression from the episomal copy retards parasite growth. Anyway, as suggested by the editor we have removed the aspect on the effect of ISN1 overexpression on parasite growth (Page 4, lines 69-70; to Page 5, lines 90-98; Page 18, lines 374-389; Fig. 1 panel D; Page 54, lines 824-836 in the original manuscript).

Perhaps fusing GFP to ISN1 affects its function and thereby reduces growth? This type of dominant effect is possible for a tetrameric enzyme.

We agree with the reviewer that dominant negative effect can be manifested as multimers of wild type ISN1 associated with wild type GFP may be formed in the cell. In a prior attempt to knock down protein levels using ISN1 tagged to GFP-DDD-HA, we did confirm that this fusion of GFP-DDD-HA to the C-terminus of ISN1 does not impair activity to any significant extent. Hence, we believe that the use of ISN1 tagged to GFP at the C-terminus to examine the effect of increased levels of ISN1 on parasite phenotype would be justified. However, again as suggested by the editor we are removing the aspect on the effect of ISN1 overexpression on parasite growth.

Perhaps the transfection procedure has affected growth? If the authors want to make this conclusion, then they firstly need to do appropriate controls, such as to over-express untagged ISN1 and see if expression levels alter. Also, they needed to produce mock transfected cells and assess expression levels.

As suggested by the editor we are deleting the in vivo growth data, hence this point is no longer relevant.

Finally, they need to set up over-expressing in a strain that produces gametocytes and to measure the number of gametocytes or conditionally knock out ISN1 and see if it leads of loss of gametocytes.

We thank the reviewer for the suggestion, we agree on this and the results from such a study will form another communication.

The authors then determine the structures for a variety of constructs. The structures mostly look well determined for their resolution. But it is hard to be sure about the quality of the crystallographic data, or whether the resolution is correct as the R factors and CC1/2 appear to be missing from the crystallographic tables? These are essential parameters, without which the quality of the data cannot be judged.

Both Rmeasure and CC1/2 values have now been added to Supplementary Table 1.

The models mostly appear good, although there are some refinement issues. Both 6MN1 and 6RMO have very high clash scores and Ramachandran outliers and these should be fixed.

We agree that these structures had high clash scores. Further refinement has been done for both 6RN1 and 6RMO structures, leading to improved clash scores and Ramachandran outliers (see new Supplementary Table 1 and data hereafter).

	6RN1	6RMO
New clash score (old ones)	6.90 (12.65)	8.38 (22.87)
New Ramachandran outliers (old ones)	0.2 (1.9)	0.15 (3.3)

Otherwise the structures appear good and are well supported by SAXS and SEC-MALLS analysis. There is also a negative stain reconstruction, which fits the crystal structure. However, a SI figure is needed for this negative stain reconstruction, showing raw data, class averages and resolution.

Okay, this has now been done (Supplementary Fig. 4) and has been inserted in the Supplementary Info.

The next section of the manuscript appears high quality and very informative, with nice figures. There is a strong set of catalytic data, identifying the substrates for this enzyme, and assessing its mechanism. There is also a very nice and detailed description of the different structural states of the enzyme in its complex with different substrates and a comparison of this with the mechanistic data. The discussion of this data, the many data tables and files and the well described methods are all praiseworthy and informative

We are happy that this referee has appreciated our work.

In summary, the strengths of this manuscript are the structural and catalytic data on an interesting enzyme expressed in human-infective Plasmodium species. This is all very well done and of high quality. The weakness of the manuscript is the claim of an important role for the over-expression of this enzyme in gametocyte development without any good evidence being presented. In my view the claims in the abstract (lines 31-35), results (lines 90-98) and conclusion (lines 366-389) are unsupported with data and should be removed. If the authors want to include these claims, they need to do the controls for Figure 1D, show the effect on gametocyte numbers of over-expression of untagged ISN1 and/or knock out ISN1 and see if it reduces gametocyte numbers. However, without these claims, this is a well-done experiment on the structure and mechanism of an interesting enzyme.

As recommended by the reviewer and the editor, these sections have been removed.

Minor points:

Line 76 What if ISN1 is present other Plasmodium species but just more divergent and hard to detect? Seems odd that it is in primate and bird infective species, but not rodent infective species? Why? Perhaps re-phrase.

A BLAST search of PlasmoDB (Aurrecochea C et al. PlasmoDB: a functional genomic database for malaria parasites. Nucleic Acids Res. 2009;37 D539-43 using PfISN1 as query picks up a total of 17 species of Plasmodium. However, all Plasmodia species that infect mice and have been sequenced, do not have ISN1. It appears that rodent malarial parasites indeed lack a sequence with homology to P. falciparum ISN1 sequence.

ISN1 belongs to the HAD superfamily of enzymes and upon a keyword search of Plasmodia database we find that these organisms have on an average 7-10 annotated HAD sequences. HAD family enzymes exhibit diverse substrate specificity and low sequence homology across members. Most of the HAD members are phosphatases and it is possible that a very divergent sequence from PfISN1 may exist in rodent malarial parasites. We have not written on this possibility in the manuscript as there is no evidence. If this referee and the editor judges that this is needed we will do so.

Line 100-101 What is the rationale for the truncations?

The rationale for the truncations was to verify and support the hypotheses observed from the structural studies, and indeed this was confirmed in that both termini are involved in regulation of the activity.

Reviewer #2 (Remarks to the Author):

Carrique L et al. present a detailed characterization of the unusual inosine monophosphate nucleotidase PfISN1 from the human malaria parasite *Plasmodium falciparum*. The manuscript content comprises 7 crystal structures, kinetic data for a variety of substrates and mutants as well as localization and overexpression studies for PfISN1-GFP in *P. falciparum* and *P. berghei*. While the structure-function analyses are quite convincing, the presentation of the kinetic data leaves room for improvement. The *in vivo* significance of the gene/protein has not been addressed properly and the drawn conclusions/hypotheses are too speculative and should be toned down or even deleted. In my opinion, the manuscript rather suffers than benefits from the *in vivo* data because the emphasis of the physiological relevance is oversold by the authors. If the authors do not want to tone this aspect down and really want to spin the story the way they do - with an emphasis on the *in vivo* significance of PfISN1 in the title and throughout the manuscript - they have to perform knock-out or CRISPR/Cas9 mutagenesis studies. This would be, in my opinion, beyond the scope of the study. I therefore rather recommend to present the highly interesting structure-function analysis as is, improve the presentation/analysis/interpretation of the kinetic data and add the localization and potential phenotype for overexpressed PfISN1-GFP as a bonus without further ado.

We agree with the reviewer that the CRISPR Cas9 mutagenesis is beyond the scope of the study. Moreover, since the editor has recommended to delete this aspect and as also stated in the answers given to reviewer 1 and suggested by reviewer 2 him-/herself, we have deleted sections dealing with these issues.

Major points:

1) Phenotype

- page 5, lines 90 - 98 and other sections: The phenotype for the PfISN1 overexpressing 3D7 strain in Fig. 1D is questionable because of a) the study design, b) the lack of a significant phenotype for 4 out of 6 data points and c) the data presentation:

a) How can the authors exclude that the growth defect did not originate from the drug treatment (WR99210) or other plasmid-derived effects that were absent in 3D7? The growth should be compared with a cell line containing the empty or a GFP(only)-encoding plasmid as a control. Furthermore, overexpression of untagged PfISN1 should be also included as a control in order to interpret the phenotype and to exclude alternative effects. At the current stage the growth defect might have many possible reasons. One of several alternative explanations that the authors neglect is a potential inactivation of PfISN1 because of a dominant-negative of the GFP tag: The authors nicely

showed that the C-terminus plays a role for the activity of tetrameric PfISN1. Hence, overexpressed PfISN1-GFP might actually interact with endogenous wild-type PfISN1 resulting in inactive heterooligomers and a decreased overall nucleotidase activity (instead of an expected increased activity in the overexpressing strain). The interpretation of the phenotype without controls with untagged (or N-terminally GFP-tagged) overexpressed PfISN1 is too speculative and the relevance of the nucleotidase for gametocyte development/drug development etc. should be toned down or even deleted.

We had examined the activity of ISN1 tagged to GFP-DDD-HA. Although ISN1-GFP-DDD-HA expressed in E. coli could not be purified to homogeneity, the activity of the partially enriched enzyme was not compromised. This suggests that tagging ISN1 at the C-terminus with GFP does not compromise ISN1 activity. We agree that a control with untagged or N-terminally GFP-tagged ISN1 may clear the doubt. However, as this part of the manuscript will be deleted, these results will be part of a future communication.

b) How do the authors explain that the growth was apparently highly similar for 4 out of 6 data

points? Maybe the difference became only relevant at a high parasitaemia that was coupled to an extreme consumption of nutrients (hypoxanthine) at day 6?

As recommended by the reviewer and the editor, this part has been removed. However, to answer the question, the data that is (was) shown in plot Fig. 1D is cumulative parasitemia normalized to 3D7; therefore 4 out of 6 points appear to overlap. Re-plot of the same data is shown in the figure below.

As seen from the same data plotted using absolute percentage values, there is a difference in growth at day 2 (6.72% vs 2.62%), day 3 (7.89% vs 3.86%) and day 4 (13.63% vs 6.51%)

Also, this experiment was performed independently four times.

c) Potential growth differences at day 1-4 cannot be seen in the graph because of the way the data is presented. Furthermore, the choice of y-axis in Fig. 1D is confusing and unusual. The experiment was started with 1% parasitaemia at day 0 and resulted in a roughly 100-fold increased parasitaemia at day 6. Taking into account a 5-fold dilution at day 4, does 100% correspond to a (potentially crashed) culture with a parasitaemia of 20%? Please plot the y-axis as % parasitaemia without normalization so that the values for days 1-4 are visible and clearly indicate the 5-fold dilution for both cultures at day 4.

As recommended by the reviewer and the editor, this part has been removed. We do understand that the way the data is presented could give rise to some confusion. However, this representation is used in many publications. By day 6, the parasitaemia has not increased 100-fold. We are taking the maximum parasitemia at day 6 as 100, and normalizing other parasitaemia values accordingly.

As is seen in the graph above, in which the y-axis is % parasitemia, the parasitemia at day 4 was around 13% for 3D7, following which the culture was diluted. At the end of day 6, the parasitemia was around 11% for 3D7 and around 5% for the 3D7_ISN1GFP strain.

We considered plotting the data by normalizing it because we felt that by doing so we would be making it more understandable to an audience which is not familiar with the way Plasmodium falciparum is cultured.

- PfISN1 was one out of 300 (!) genes that were upregulated in gametocytes in the study by Brancucci et al. A slight growth reduction of asexual blood-stage parasites that overexpressed PfISN1-GFP was observed in cell culture experiments without adequate controls. These two facts combined are not sufficient for a proof-of-principle or the generalized conclusions regarding the physiological significance of PfISN1 for gametocyte development. If the authors really want to make this extremely speculative connection (as emphasized in the summary and throughout the manuscript) they should a) determine the concentrations of IMP/AMP/GMP for strains with an empty plasmid as well as overexpressed PfISN1-GFP vs. overexpressed untagged PfISN1, b) quantify PfISN1 in these strains by western blotting using their antibody and various concentrations of recombinant PfISN1 for calibration, and c) quantify whether *P. falciparum* and *P. berghei* parasites that overexpress PfISN1 form more/less female/male gametocytes. If the authors are right, the purine nucleotide concentrations should be drastically altered depending on the PfISN1 concentration (as hypothesized by the authors, page 5, lines 94-98) and *P. falciparum* and *P. berghei* with overexpressed PfISN1 should form more/less female/male gametocytes.

We thank the reviewer for the suggestions, and indeed many of these experiments suggested by this reviewer are being carried out in the laboratory. Cf. earlier responses to comments to this manuscript and suggestions by the editor and reviewers, we believe that for this manuscript, these suggestions are outside the scope of the study.

- page 4, lines 76-81 and many other sections: If ISN1 is such a central metabolic enzyme, why was it presumably lost in all rodent malaria parasites? Furthermore, a check of the piggyBac insertion mutagenesis data in PlasmoDB suggests that PfISN1 is dispensable and that its loss has no fitness cost. Please cite the study (Zhang et al 2018 Science 360:6388) and take it into account regarding the central hypothesis on the relevance of PfISN1.

*It is indeed very interesting that of all species of plasmodia whose genomes have been sequenced, ISN1 is absent only in the murine malarial parasite *P. berghei*, *P. yoeli*, *P. chabaudi* and *P. vinckei*. It is present in 18 species of plasmodia that infect human, primates and birds. The significance of the loss of ISN1 in rodent malarial parasites is not clear at this stage. The study by Brancucci et al. shows that commitment to sexual stage development is mediated by different factors in the human malarial parasite *P. falciparum* and in the mouse malarial parasite *P. berghei*. The absence of ISN1 in *P. berghei* probably reflects the different mechanisms by which commitment to sexual stages takes place. We are indeed aware of the fact that ISN1 is not essential for asexual development and its increased expression is seen in female gametocyte. At this stage it can only be said that this is one of the genes that plays a role in female gametocyte development. Further studies are needed to understand the exact role of this enzyme. Zhang et al 2018 Science 360:6388 is now cited in the revised manuscript p. 4 lines 68-70. We have also added the highlighted sentence in the section on page 4: "Gametocyte formation brought about by LysoPC depletion was associated with activation of expression of more than 300 genes, including genes involved in phosphocholine (PC) biosynthesis, DNA replication and macromolecule modification. Interestingly, ISN1 is also strongly induced. However, genome wide disruption in *P. falciparum* by piggyBac transposon insertion suggests that ISN1 is mutable in the asexual stages without loss of fitness (Zhang et al 2018 Science 360:6388)."*

2) Kinetics

- Whole manuscript: The terms kcat, Km, kcat/km, Kd etc. are incorrect because all values reflect apparent (and not true) kinetic parameters that were derived from primary plots at one constant parameter (e.g. one constant ATP and/or Mg²⁺ concentration). If the authors want to make claims about the true kinetic constants of the enzymes, they have to extrapolate the constants (from secondary plots) based on systematic variations of all relevant parameters. This would be quite some work. Thus, I rather recommend to replace all terms with apparent kcat, apparent Km etc. and to be much more careful about the interpretation of the constants (e.g., regarding substrate affinities under physiological conditions, the assignment of pH activity profiles and a macroscopic pKa value to different ionisable groups in the substrate and enzyme etc.).

Okay, as requested by the reviewer we have changed K_m and k_{cat} to apparent K_m and apparent k_{cat} .

- Extended Data Fig. 4 and page 7: This part of the manuscript lacks clarity and is difficult to follow because of the incorrect figure legend. Poor data fits in panels B (pKa extrapolation), panel C ([IMP] variation) and panel E ([Mg²⁺] variation) raise doubts about the used equations, kinetic models and interpretations for all indicated parameters (sumarized in Table 1). The kinetic data of the tetrameric enzyme are obviously too complex to be fitted using a simple M.-M. or Hill equation. Alternative equations, models etc. that might result in better fits can be found for the pKa values in Brocklehurst K 1996 (Physical factors affecting enzyme activity. In: Enzymology Labfax 175–198) and for oligomeric proteins with modulators in Segel IH 1993 (Enzyme Kinetics: Behavior and Analysis of Rapid Equilibrium and Steady State Enzyme Systems). Please repeat the analyses with appropriate equations that are based on plausible kinetic models and correct the according paragraphs.

We apologize for the incorrect legend. The modified legend in the revised draft is pasted below. Please notice that due to demands from the referees of inserting further information in the Extended data section this figure has now become Supplementary Fig. 6 in the revised manuscript.

Supplementary Fig 6: Kinetic characterization of PflSN1. (A)pH-dependence of ISN1 activity. Inset shows stability of PflSN1 as a function of pH. Substrate used in both cases was IMP. (B)Log (k_{cat}/K_m)_(app) vs pH. Line represents fit of the data to eq. 4. Inset shows log ($k_{cat(app)}$) vs pH. Substrate used was IMP. (C)Initial rate vs [IMP] at pH 8.0 and pH 5.0 (inset). Line represents fit of the data to eq. 1. (D)Initial rate vs [AMP] at pH 8.0 and pH 5.0 (inset). Lines represent fit of the data to eq. 3 (main panel) and eq. 1 (inset). (E)Initial rate vs [MgCl₂], with IMP as substrate at pH 8.0 and pH 5.0 (inset). The lines represent fit of the data to eq. 3 (main panel) and eq.1 (inset). (F)Initial rate vs [pNPP] plot at pH 8.0 with 30 mM Mg²⁺. The line represents fit of the data to eq. 3. All assays were performed in duplicate and repeated at least three times. The plots shown correspond to one of the replicates. The values in the plots are mean with error bar denoting standard error of the mean. v , initial rate. The unit of v is μmol of product formed in one sec. Eq. 1-3 are described in Methods. Details of equations used to select the model that best fits the data are provided in the Methods section. BELL equation was the preferred model for $\log((k_{cat}/K_m))_{(app)}$ vs pH plot in panel B. Hill equation was the preferred model for main panels of D and E and panel F. Plots in panel C, insets in panel D and E were fit to the Michaelis-Menten equation.

Poor data fits in panels B (pKa extrapolation), panel C ([IMP] variation) and panel E ([Mg²⁺] variation) raise doubts about the used equations, kinetic models and interpretations for all indicated parameters (sumarized in Table 1).

a) Poor data fits in panels B (pKa extrapolation)

Panel B was fitted to four different equations (see section on ‘Analysis of kinetic data’ in Methods) representing different ionization models. Best-fit was seen with BELL equation ($k_{cat}/K_m = c/(1+H/K_1+K_2/H)$). Panel B (now in Supplementray Fig. 6) has been replaced with the line representing fit to BELL equation. We thank the reviewer for pointing this out.

b) panel C ([IMP] variation)

Please notice that at pH 8.0, the $K_m(\text{app})$ value for IMP is very high. The equation used for both the main panel and the inset is the Michaelis-Menten equation (Eq. 1). The $V_{\text{max}}(\text{app})$ value obtained from the fit is 15.3 ± 0.9 at pH 8.0 and 11.6 ± 2.1 at pH 5.0. The $K_m(\text{app})$ value obtained from the fit is 66.0 ± 6.3 mM at pH 8.0 and at 0.34 ± 0.09 mM pH 5.0. Here again, the error values are low. It should be noted that the primary aim of showing these data is that the $K_m(\text{app})$ value for the enzyme at pH 8.0 is very high while at pH 5.0 a dramatic drop is seen.

The output from the fits are given below

IMP pH 8.0:

Best-fit values

Vmax	13.51
Km	65.80
Std. Error	
Vmax	0.3957
Km	4.125

95% Confidence Intervals

Vmax	12.69 to 14.33
Km	57.24 to 74.35

IMP pH 5.0

Best-fit values

VMAX	9.600
KM	0.3073
Std. Error	
VMAX	0.5306
KM	0.04691

95% Confidence Intervals

VMAX	8.489 to 10.71
KM	0.2091 to 0.4055

c) panel E ([Mg²⁺] variation)

The equations used are the Hill equation for the main panel and Michaelis-Menten equation for the inset. We refer in the manuscript to this figure to highlight that v v/s Mg²⁺ plots show sigmoidal behaviour at pH 8.0 that becomes hyperbolic at pH 5.0. The key purpose of including this figure is to highlight the presence of some level of cooperativity in the enzyme. The results from fit to Hill and Michaelis-Menten equation are provided below. Fit to KNF model was also ambiguous

v v/s Mg plot

Comparison of Fits

Null hypothesis	Michaelis-Menten
Alternative hypothesis	Hill equation
P value	P<0.0001
Conclusion (alpha = 0.05)	Reject null hypothesis
Preferred model	Hill equation
F (DFn, DFd)	42.62 (1,24)

Hill equation

Best-fit values

VMAX	14.07
H	1.713
KM	26.64

Std. Error

VMAX	0.4531
H	0.1267
KM	1.602

95% Confidence Intervals

VMAX	13.14 to 15.01
H	1.452 to 1.975
KM	23.33 to 29.94

Goodness of Fit

Degrees of Freedom	24
R ²	0.9903
Absolute Sum of Squares	5.710

Michaelis-Menten

Best-fit values

VMAX 19.69

KM 53.14

Std. Error

VMAX 1.161

KM 6.768

95% Confidence Intervals

VMAX 17.30 to 22.09

KM 39.19 to 67.08

Goodness of Fit

Degrees of Freedom 25

R² 0.9730

Absolute Sum of Squares 15.85

- d) The kinetic data of the tetrameric enzyme are obviously too complex to be fitted using a simple M.-M. or Hill equation.

The plots showing sigmoidal behavior were fit to Hill and MWC equation as described in the section on 'Analysis of kinetic data' in Methods. In all cases only fit to Hill equation gave statistically significant values for the kinetic parameters. Results from the fits are given below

Comparative test: Akaike's Information Criteria (AICc)

v vs Mg plot

Comparison of Fits	Can't calculate
Simpler model	MONOD SIMPLE

Probability it is correct	Ambiguous
Alternative model	Hill equation
Probability it is correct	
Ratio of probabilities	
Preferred model	Hill equation
Difference in AICc	
MONOD SIMPLE	Ambiguous
Best-fit values	
VMAX	12.13
KR	~ 0.004185
L	~ 7.556e+014
Std. Error	
VMAX	0.3997
KR	~ 6.257e+006
L	~ 4.519e+024
95% Confidence Intervals	
VMAX	11.31 to 12.96
KR	(Very wide)
L	(Very wide)
Goodness of Fit	
Degrees of Freedom	24
R ²	0.9402
Absolute Sum of Squares	35.04
Hill equation	
Best-fit values	
VMAX	14.07

H	1.713
KM	26.64
Std. Error	
VMAX	0.4531
H	0.1267
KM	1.602
95% Confidence Intervals	
VMAX	13.14 to 15.01
H	1.452 to 1.975
KM	23.33 to 29.94
Goodness of Fit	
Degrees of Freedom	24
R ²	0.9903
Absolute Sum of Squares	5.710

v vs AMP plot

Comparison of Fits	Can't calculate
Simpler model	MONOD SIMPLE
Probability it is correct	Ambiguous
Alternative model	Hill equation
Probability it is correct	
Ratio of probabilities	
Preferred model	Hill equation
Difference in AICc	
MONOD SIMPLE	Ambiguous
Best-fit values	

VMAX	14.25
KR	~ 1.890
L	~ 3.019e+006
Std. Error	
VMAX	0.8345
KR	~ 2715
L	~ 1.735e+010
95% Confidence Intervals	
VMAX	12.11 to 16.40
KR	(Very wide)
L	(Very wide)
Goodness of Fit	
Degrees of Freedom	5
R ²	0.9930
Absolute Sum of Squares	1.159
Sy.x	0.4814
Hill equation	
Best-fit values	
VMAX	12.51
H	5.311
KM	72.70
Std. Error	
VMAX	0.3447
H	0.3219
KM	1.294
95% Confidence Intervals	
VMAX	11.62 to 13.39
H	4.483 to 6.138
KM	69.37 to 76.03
Goodness of Fit	

Degrees of Freedom	5
R ²	0.9987
Absolute Sum of Squares	0.2124

v vs PNPP plot

Comparison of Fits	Can't calculate
Simpler model	MONOD SIMPLE
Probability it is correct	Ambiguous
Alternative model	Hill equation
Probability it is correct	
Ratio of probabilities	
Preferred model	Hill equation
Difference in AICc	

MONOD SIMPLE	Ambiguous
Best-fit values	
VMAX	0.002560
KR	~ 0.01030
L	~ 1.028e+011
Std. Error	
VMAX	5.164e-005
KR	~ 65272
L	~ 2.605e+018

95% Confidence Intervals

VMAX	0.002452 to 0.002669
KR	(Very wide)
L	(Very wide)

Goodness of Fit

Degrees of Freedom	18
R ²	0.9668
Absolute Sum of Squares	4.775e-007

Sy.x	0.0001629
Hill equation	
Best-fit values	
VMAX	0.002664
H	2.952
KM	5.937
Std. Error	
VMAX	5.474e-005
H	0.2388
KM	0.1863
95% Confidence Intervals	
VMAX	0.002549 to 0.002779
H	2.451 to 3.454
KM	5.545 to 6.328
Goodness of Fit	
Degrees of Freedom	18
R ²	0.9820
Absolute Sum of Squares	2.590e-007

- What are the expected physiological concentrations of ATP and IMP in *P. falciparum* and how are these concentrations reflected by the chosen assay conditions and sigmoidal curves in primary plots?

To the best of our knowledge, there is no single study that measures both ATP and IMP concentrations in the parasite. The report on concentrations of purines and pyrimidine nucleotides in the parasite gives a number for IMP while it does not quantify ATP (Laourdakis CD, Merino EF, Neilson AP, Cassera MB. Comprehensive quantitative analysis of purines and pyrimidines in the human malaria parasite using ion-pairing ultra-performance liquid chromatography-mass spectrometry. J Chromatogram B Analyst Techno Biomed Life Sci. 2014, 967:127-33). Metabolite profiling by NMR quantifies ATP but does not provide a number for IMP (Tang R, Junankar PR, Bubb WA, Rae C, Mercier P, Kirk K. Metabolite profiling of the intraerythrocytic malaria parasite Plasmodium falciparum by (1)H NMR spectroscopy. NMR Biomed. 2009, 22, 292-302). Other papers do report levels of ATP (Fry M, Webb E, Pudney M. Effect of mitochondrial inhibitors on adenosine triphosphate levels in Plasmodium

falciparum. *Comp BiochemPhysiol B*. 1990, 96, 775-82). Further the units used across these papers also vary. Our attempts at normalization of these numbers to a common unit show that there is 10-100 fold variation across the studies (0.225-2 mM). IMP concentration has been estimated by one study and this is reported as 7 nmol/10⁷ cells, which upon conversion is 25 mM. The concentrations of ATP and IMP in the RBC as reported by Lourdakis et al is 175 mM and 9.7 mM, respectively (this is after conversion of the numbers provided to molarity). A comparative analysis of metabolite concentration in uninfected and *P. berghei* infected erythrocytes by LC-MS, indicate that IMP level are elevated by 8.7-fold upon infection whereas the value for ATP is not given. Further, this study mentions only fold increase and does not quantify the levels (Anubhav Srivastava, Comparative metabolomics of erythroid lineage and Plasmodium life stages reveal novel host and parasite metabolism, 2014, Ph. D thesis, University of Glasgow, Host-parasite Interactions Revealed by Plasmodium falciparum Metabolomics-Cell Host Microbe. 2009, 5, 191-9). With the lack of necessary information available, it is difficult for us to provide a suitable answer to the question. The concentration of ATP used to examine the effect on IMP hydrolysis was 4 mM. This concentration is within the values reported.

- Table 1: The primary plots and data fits for most of the data are not shown in the manuscript and should be included.

Okay, we have added the missing plots in the Supplementary info (Supplementary Fig. 7).

- Table 1 and all figures showing kinetic data: The exact conditions should be indicated (e.g., modulator concentration tested, enzyme concentration, temperature, salt concentrations, buffer concentrations etc.) for each set of experiments because the apparent constants can highly depend on them.

Sorry for this, we have rewritten the Methods and also included details in footnote to Table 1 and in the legends to figures.

- Extended Data Fig. 6: Please indicate which type of inhibition the plot and fit are based on, why this type of inhibition was chosen and why the inhibition constant is interpreted as an apparent K_d value.

This experiment examines the binding of IMP to the inactive mutants D170N and D172N of PfISN1. D170 and D172 are residues present in motif 1 of ISN1. In the HAD family enzymes, these two aspartyl residues in motif 1 are involved in catalysis. Though these mutants were inactive on IMP, they were active on pNPP. Further we also observed that the presence of IMP decreased the activity on pNPP. Through inhibition of this activity on pNPP, we measure the affinity of the mutants for IMP. This "guided us" in our crystallization experiments of PfISN1_{D172N} in complex with IMP. The main aim of doing this was to get an idea of the affinity of the mutants for IMP before obtaining ligand bound crystals. The binding of IMP to PfISN1_{D172A} and PfISN1_{D172N} monitored as inhibition of pNPP hydrolysis activity was fit to one site binding equation,

$$\% \text{ inhibition of } v = (B_{max} \times I) / (K_d + I) \quad \text{eq. 7}$$

where % inhibition of $v = 100 - (v_i/v_0)$ (v_i is the enzyme activity at various IMP concentrations and v_0 is the enzyme activity at 0 mM IMP), I is the inhibitor (IMP) concentration, B_{max} is the maximum % inhibition of v , and binding constant K_d is the inhibitor concentration at which half-maximum inhibition is achieved.

We have mentioned the equation in the section 'Analysis of kinetic data' in the Methods part of the revised manuscript.

- Page 9 and Extended Data Fig. 7: The data in Fig. 7 does not show that ATP is a competitive (K-type) inhibitor. Please show the according M.-M. and Lineweaver-Burk plots for different IMP and AMP concentrations to support the statements on the apparent K_m and k_{cat} (V_{max}) values. Please indicate which equation/model/fit was used for the plot in Fig. 7B and how the "affinity" was determined.

Please, note that ATP is not an inhibitor but an activator. The binding of ATP leads to 10 fold drop in the K_m value for IMP from 66.6 to 6.8 mM while the increase in V_{max} is small. Compounds that activate an enzyme by increasing the affinity (lowering the value of K_m) for the substrate without significantly affecting V_{max} are referred to as K-type activators. Our studies show that ATP is a K-type activator of the IMP hydrolyzing activity of ISN1. Assays monitoring IMP hydrolysis was done in the presence of 4 mM ATP.

The plot (now Supplementary Fig. 7B) was fitted to one-site binding equation and this is written in the legend to the figure. The equation is

$$v = (C_{max} \times A)/(K_d + A)$$

Where v is fold activation, A is the activator (ATP) concentration, C_{max} is the maximum fold activation, and binding constant K_d is the activator concentration at which half-maximum activation is achieved. These details are now provided in the section 'Analysis of kinetic data' in Methods.

- Page 10, line 203-205; Fig. 3 shows an activity plot. No data on the apparent catalytic efficiencies are shown.

Fig 3 shows Specific Activity of various mutants as compared to the wild type enzyme under similar conditions. Table 1 lists the apparent catalytic efficiencies for $\Delta N30$, $\Delta C10$, K41L and H398V for the substrate IMP. Catalytic efficiencies were determined for D172N and D172A with pNPP as substrate. The mutants D170N, D172N, D172A, D170N-D172N, Y176L, R218L, D394V, Q395L, F396L, D402V, R406L, W413L, $\Delta N59$, $\Delta N30$ - $\Delta C10$, D363V, W365L, and D367V are all inactive on IMP as substrate.

Mutants D178V, F403A, W365Y, and W365F showed weak activity with IMP as substrate. For all these the catalytic efficiencies could not be measured. The mutants H150V and H150V- $\Delta C10$ were generated to examine the role of these residues in mediating the effect of ATP binding. The main purpose of making the mutants was to make correlations with structural observations. As expected from structural observations, $\Delta C10$ shows no activation by ATP.

F403 shows different side chain conformations across the structures and the mutants studied indicate that the aromatic nature of this residue is essential for wild type phenotype.

3) Others

- The PDB reports for structures 6RMO, 6RN1 and 6RMW show numerous clashes and Ramachandran outliers. Please clarify.

We agree, and further refinements have been done for 6RN1, 6RMO and 6RMW structures, resulting in improved refinement statistics, notably as concerns clash scores and Ramachandran outliers (see new Supplementary Table 1 and data below). The limited improvement of 6RMW is due to the quality of the data, which however is good enough to for verifying the overall fold and hypotheses set forward in the manuscript.

	6RN1	6RMO	6RMW
New clash score (old ones)	6.90 (12.65)	8.38 (22.87)	9.06 (9.62)
New Ramachandran outliers (old ones)	0.15 (1.89)	0.15 (3.3)	0.27 (0.44)

- Extended Data Fig. 11B: Poor data fit. Does the fit take into account potential multiple binding sites/allosteric/cooperative effects for each subunit of the homodimer? There is no insert in contrast to the figure legend. Please clarify.

Thanks for noticing. The fit does not take cooperativity into consideration. The best fit was obtained with single site binding model. The binding stoichiometry was obtained from this fit.

The sentence "Best-fit parameters for one-site binding model are shown in inset." has been inadvertently introduced. This sentence is deleted in the revised version of the manuscript.

Minor points:

- page 2: Some parts of the summary are too vague or imprecise and do not really describe the results. The first sentence has nothing to do with the study and could be deleted. The statement that the structures "reveal complex domain organizations tightly associated with catalysis" could be more specific. The *in vivo* overexpression data is limited to PfINS1-GFP (not PfINSP) and the hypothesis on the metabolic switching at the end of the summary is too speculative considering the available data and the absence of ISN1 in numerous Plasmodium species.

The first sentence relates to the "background" of the study which is demanded from the journal; the sentence "reveal complex..." has now been changed to "reveal complex rearrangements of domain organization tightly associated with catalysis".

- page 2, line 27: Please introduce the abbreviation ISN1.

ISN1 has been introduced.

- page 3, line 55: Maybe replace "homology" with "similarity" (two genes/proteins are, as far as I understand the concept, either homologous or not, but genes/proteins can have different degrees of similarity).

Yes indeed, we thank the reviewer for pointing out this discordance. We have replaced homology with similarity.

- References are missing to support the statements. For example:
page 3, line 42: "Purine nucleotides are mandatory... phosphotransferase activity." (4 sentences without reference)

We thank the referee for noticing and apologize for the oversight; the following references have been inserted/included:

Purine and Pyrimidine Pathways as Targets in Plasmodium falciparum

María Belén Cassera, Yong Zhang, Keith Z. Hazleton, and Vern L. Schramm Curr Top Med Chem. 2011, 11, 2103–2115 (article was referred to elsewhere in the original submission)

Mammalian 5'-nucleotidases. Bianchi V, Spsychala J. J Biol Chem. 2003 Nov

21;278(47):46195-8. (newly inserted)

The

bifunctional cytosolic 5'-nucleotidase: regulation of the phosphotransferase and

nucleotidase activities. Pesi R, Turriani M, Allegrini S, Scolozzi C, Camici M, Ipata PL, Tozzi MG. Arch BiochemBiophys. 1994, 312, 75-80 (article was referred to elsewhere in the original submission)

page 3, line 56: "Plasmodia lack... including humans."

There is no paper that specifically demonstrates the absence of the cN II class of 5' purine nucleotidase in Plasmodium. In the annotated database "PlasmoDB" a keyword search for cN II class of 5' purine nucleotidase does not yield any hits.

In addition a BLAST search of the NCBI database for homologs of eukaryotic or prokaryotic cN II does not yield any hits with significant sequence similarity in Plasmodium. The sequence identity between human and parasite enzymes is 11%.

- page 5, line 86: Please indicate that the antibody was generated in this study against recombinant PfISN1.

The supplementary section provides the protocol used for PfISN1 antibody generation and purification using an antigen affinity column. We have now indicated in the legend to Fig. 1 that the antibody against PfISN1 was generated in this study: "The antibody used was generated as described in Methods"

- Please include quality controls for the new antibody (e.g., a western blot as extended Fig. 1D and a negative control in Fig. 1A).

A Western blot that shows specific binding of the antibody to only the recombinant protein has been included. Controls including no primary antibody and preimmune serum for immunofluorescence microscopy are now included in the Supplementary info (Supplementary Fig. 2).

- page 25, line 571: Please replace "episomal expression of ISN1" with "episomal expression of PfISN1-GFP" and indicate how the growth assay was started (synchronous ring-stage parasites at an initial parasitaemia of 1%?)

This aspect of the study has been removed as recommended by the editor.

- Please explain to the unexperienced reader whether the PbEf1alpha promoter is a strong/weak constitutive/stage-dependent promoter in *P. falciparum*.

The PbEf1alpha promoter is a strong constitutive promoter with high level of expression in all the asexual intraerythrocytic stages with trophozoites showing higher level of expression. This statement is now included in Methods (P. 33).

- The upper row in Fig 1B shows a trophozoite (not a ring stage) parasite.

Sorry, the stages indicated are only for Fig. 1A. This is now stated in the legend.

- Page 5, line 104: Is there a reference for the domain nomenclature? If yes, please cite the reference(s).

To the best of our knowledge, there is no reference for the domain nomenclature "oligomerisation domain" (OD), "N-Terminal Regulation Domain" (NTRD), whereas for the Cap domain this has been described earlier in the HAD family of enzymes. We have inserted a reference to this now (K. Allen & D. Dunaway-Mariano, 2004) and apologize for having omitted this. Finally, we believe that none of the earlier studies of the HAD family of enzymes talks about the "catalytic domain" (CD) but rather a core domain. Since this domain seems

rather different from the “core domains” described in the literature, we have chosen to call it “Catalytic Domain”. Should you have some counter-examples that we have overseen we will of-course insert the references accordingly.

- Extended Data Fig. 2: Please include data for a calibration curve, add the calculated molecular mass for monomeric recombinant PfISN1 to the figure legend and indicate the apparent molecular mass for the peaks at roughly 10.2 and 11.9 mL.

The figure has been modified (Supplementary Fig. 3) and completed as suggested by the reviewer.

- page 7, line 129 and other sections, y-axis labels in Extended Data Fig. 4+8 etc.: Please do not use $\mu\text{mol substrate per min \& mg enzyme}$ ("umol min⁻¹ mg⁻¹") as units. The y-axis label "v" is incorrect. Please use the correct axis label " $v/[\text{enzyme xy}]$ ". The correct unit is "sec⁻¹".

The y-axis labels have been changed to “v/[E]” with units as sec⁻¹.

- page 8, line 150-152: Reference supporting statement missing.

The following reference has been added: A. Maxwell Burroughs, Karen N. Allen, Debra Dunaway-Mariano, L. Aravind. J. Mol. Biol. 361, 2006, 1003-1034. Evolutionary Genomics of the HAD Superfamily: Understanding the Structural Adaptations and Catalytic Diversity in a Superfamily of Phosphoesterases and Allied Enzymes.

- page 8, line 150-152: "on the synthetic compound pNPP, D172N and D172A mutants showed 150- and 25-fold increased activity". Where is this data shown?

The fold increase in activity is 91 and 15, respectively. We apologize for the error. The figure is provided as Panel A in Supplementary Fig. 9, and corrections have been made accordingly in the text.

- page 9 and Extended Data Fig. 8: Please indicate the conditions that lead to 79/87% inhibition.

The assay conditions are described in the methods section of the revised draft. The following changes have been made to improve clarity. “To assess the magnitude of inhibition of PfISN1 activity by phosphocholine and myo-inositol-4-phosphate (Sigma -Aldrich), activity was measured in 50 mM Tris- HCl, pH 8.0, 30 mM MgCl₂ and 5 mM IMP. The concentration of the inhibitor used was 2 mM and the assay mix containing the enzyme was incubated for 3 minutes at 25°C. The released phosphate was estimated as described above. Control reaction had the substrate and the inhibitor but lacked the enzyme”.

- page 59, line 932: Do the duplicates represent independent expression and purification experiments (biological replicates) ?

The wild type had more than three biological replicates. Most mutants were also purified more than once and enzyme assays were done.

- Extended Data Fig. 14: Alevolata (typo)

Thanks, corrected

- Fig. 5: The panels in the upper row are too small. Maybe use a 3x2 instead of a 2x3 format?

OK. The figure has been modified as suggested by the reviewer.

Reviewers' comments:

Reviewer #1 (Remarks to the Author):

The authors have responded positively to my comments and I believe that the manuscript is much improved. The unconvincing data has been removed and the statistics associated with the models are now much better.

The authors have also now included numbers of CC1/2 and Rmeas. These have highlighted difficulties with some of the data sets.

The structures for APO, ATP and deltaN59 all appear reasonable.

However, the D172N-IMP and the deltaC10 have very low CC1/2 values in the outer shell and very high Rmeas. I would suggest that this means that the resolution is not as high as suggested by the authors. I would recommend them to re-process the data with a resolution cut-off that gives a CC1/2 in the region of 0.5 in the outer shell and see if the Rmeas is improved.

The D172N-deltaN30-IMP is a bit more confusing as the CC1/2 looks OK and but the Rmeas is very high. What do the authors think is the reason for this and what have they done to improve these statistics?

In my view, this should be addressed, as the resolution of the data must be appropriate for the biological conclusions that are drawn. The placement of the secondary structural elements will no doubt be right at these higher resolutions, making description of conformational changes reliable, but the side chain interactions described are less reliable at these interactions. Perhaps the authors should also give electron density maps for key regions in these lower resolution maps?

Reviewer #2 (Remarks to the Author):

The revised manuscript by Carrique L et al. is much improved and addressed the main point that was raised by both referees and the editor (regarding the lack of data to support the physiological relevance of PfISN1). Most of the other points were also addressed in the revised manuscript. Several mediocre data presentations/figures in the supplements still distract from the interesting data and some minor errors and imprecisions still remain to be corrected before publication. For example:

Throughout the whole manuscript and supplements: "The unit of v is μmol of product formed in one sec." The statement is wrong and should be corrected. The y-axis of the Michaelis-Menten plots now shows " $v/[\text{enzyme}]$ " with the correct unit " sec^{-1} " with v as a change of concentration (not an amount of substance!) over time (unit " $\text{M}/(\text{M} \times \text{sec}) = \text{sec}^{-1}$ "). The authors apparently mix up the definition of an enzyme unit (conversion of μmol substrate per min) and the definition of a reaction velocity. It is interesting to note that 11 authors publish a manuscript with half of its data on enzyme kinetics and none of the authors recognizes this very basic error in the legends of Fig. 3, Fig. S6, Fig. S7, Fig. S11 as well as the methods section. Or was there a general mistake regarding the data analysis?

Supplementary Fig.1: "B)List of all Plasmodium species containing ISN1 genes." Should be "List of all currently sequenced Plasmodium species containing ..."

Supplementary Fig. 3: Please do not use the term "weight" in combination with the unit kDa. Use the term "mass" instead (3 times in the legend).

Supplementary Fig. 6: The insets are hardly readable. Please improve the quality of the figure according to the standards of the journal.

Supplementary Fig. 9: K_d in panel B should be apparent K_d .

Supplementary Fig. 10: The figure includes blurry screen shots and a small inset. Please improve the quality of the figure according to the standards of the journal.

Supplementary Fig. 11: There are confusing background numbers in panel B (which is not labelled) that might have been added manually. Please correct the figure.

Supplementary Fig. 14: The data fit is still poor with a systematic distribution of data points above (ratio 0.3 to 0.7) and below (ratio 0.6 to 1.6) the fit. A likely explanation for the observed discrepancy is the selection of a static single site binding model. Even though the stoichiometry of 1:1 might not change with better models/fits that take into account a potential cooperativity of the oligomeric protein, the authors appear to miss a good opportunity to further support their statement on effector-dependent domain rearrangements that are relevant for catalysis (e.g., IMP-induced fit of a closed conformation).

Page 6, line 122: At a concentration of 10 mM, only IMP and, to a much lower extent, AMP and p-nitrophenyl phosphate (pNPP, non-physiological substrate) were found to be substrates with specific activity values of 3.2 ± 0.5 , 0.07 ± 0.03 and 0.002 ± 0.0002 sec⁻¹, respectively. (I tried to correct multiple errors in this sentence and hope that it makes more sense now).

Page 6, line 128: "whereas the K_m value dropped" should be "apparent K_m ".

Page 7, line 131: "drop in K_m suggests" should be "apparent K_m ".

Page 7, line 133: "The pK_1 and pK_2 values of 4.95 ... reflects that both substrate ionization and an acid group on the enzyme are involved in binding" should be "The apparent pK_1 and pK_2 values of 4.95 ... might reflect that both substrate ionization and an acid group of the enzyme are involved in binding." If the authors insist on their wording they would have to show controls with $\log(K_{cat(app)}/K_m(app))$ plots for mutants and variable charged substrate analogues to support the statement.

Page 10, line 201: "higher catalytic efficiency (k_{cat}/K_m)" should be "higher catalytic efficiency ($k_{cat(app)}/K_m(app)$)".

Legend Fig. 2: "orientation as seen in figures 3C and 3D are indicated." Should be "2C and 2D"

Legend Fig. 2: "behind subunits b (left-hand side) and a (right-hand" ... "an NTRD from subunit a" ... "two OD's of subunits β and d." Please use greek symbols for all subunits.

Abstract and Discussion: "With earlier evidence on *isn1* upregulation in female gametocytes, the structures reported in this study form the basis for new transmission-blocking agents." and "The presence of ISN1 in the plasmodial species causing malaria in humans and its absence in the host raises the possibility of developing therapeutics targeted through this enzyme." This referee still thinks that the therapeutic aspect is rather far-fetched but leaves it up to the editor whether the two statements make sense or not.

Responses to reviewers (please notice that due to the changes that have been made, the original numbering given in questions from reviewers is no longer exactly the same)

Reviewers' comments

Before answering the remaining questions we would like to thank the reviewers for having read this manuscript in a really detailed manner as well as their suggestions to improve the manuscript.

Reviewer #1 (Remarks to the Author):

The authors have responded positively to my comments and I believe that the manuscript is much improved. The unconvincing data has been removed and the statistics associated with the models are now much better.

We are happy to hear that this referee finds the manuscript much improved and thank for the suggestions.

The authors have also now included numbers of CC1/2 and Rmeas. These have highlighted difficulties with some of the data sets.

The structures for APO, ATP and deltaN59 all appear reasonable.

However, the D172N-IMP and the deltaC10 have very low CC1/2 values in the outer shell and very high Rmeas. I would suggest that this means that the resolution is not as high as suggested by the authors. I would recommend them to re-process the data with a resolution cut-off that gives a CC1/2 in the region of 0.5 in the outer shell and see if the Rmeas is improved.

We entirely agree with this referee that the data as suggested by CC1/2 and Rmeas values are not really at a resolution as high as what is given. In fact we already did process them at lower resolutions at the time where they were collected. However, since the resulting electron density differences were marginal, we at that time decided to cut them at a resolution which is higher

than what we normally would have done. Indeed, we have tested a really large number of crystals before obtaining the data that are analyzed in this manuscript. But as suggested by this referee we have reprocessed the data in order to have better statistics for these two values. As can be seen in Supplementary Table1, these values have been improved for D172N-IMP. We have taken the opportunity to do the same with the data of ATP bound PfISN1 data that also had a CC1/2 value in the outermost shell which was limited and idem for Rmeas. Finally we really are very sorry but some of the parameters for the deltaC10 structure given in Supplementary Table 1 came from a refinement of an earlier structure (values have been replaced in Supp Table 1 and are highlighted in yellow) and which for an unknown reason had not been replaced. Therefore these data have not been reprocessed, and we hope that the referee and editor agree with us in not doing so. We really apologize on this and have triple checked that there is no such problem elsewhere in the table. We would like to emphasize that differences in the 3D structure of the newly refined structures are really marginal, so all interpretations given in the manuscript still are valid.

Structure-ID	PfISN1_SeMet-ATP	PfISN1-Apo	PfISN1-ATP	PfISN1 _{D172N} -IMP	PfISN1-ΔC10	PfISN1 _{D172N} -ΔN30-IMP	PfISN1-ΔN59
Data collection							
Beamline (ESRF)	ID29	ID23-1	ID30A	ID23-2	ID30B	ID30B	ID30-A3
Wavelength (Å)	0.979	0.973	0.967	0.872	0.976	1.000	0.967
Space group	P 6 ₅ 22	P 2 ₁	P 6 ₅ 22	P 2 ₁	P 6 ₅ 22	P 2 ₁	P 4 ₂ 2 ₁ 2
Cell dimensions							
a, b, c (Å)	210.2 210.2 105.8	149.0 204.1 149.2	210.7 210.7 106.2	108.3 204.6 115.9	211.9 211.9 105.1	107.8 203.9 115.3	140.7 140.7 87.0
α, β, γ (°)	90 90 120	90 90.02 90	90 90 120	90 113.2 90	90 90 120	90 113.4 90	90 90 90
Resolution range (Å)	47.1 - 3.7	48.3 - 2.6	47.4 - 2.8	47.7 - 3.4	19.9 - 3.7	49.7 - 3.5	29.6 - 3.0
Total reflections	590962	668774	607927	228074	288982	369099	492245
Unique reflections	27949	250293	34615	63502	15185	57015	18036
R _{meas} (%)	14.5 (94.3)	11.8 (69.4)	10.2 (144.5)	18.0 (101.6)	40.4 (181.3)	39.2 (128.6)	14 (91.1)
CC _{1/2} (%)	99.9 (93.5)	99.6 (81.9)	100.0 (77.0)	99.4 (54.8)	99.8 (79.3)	99.4 (76.7)	99.9 (95.6)
I/σ(I)	21.1 (5.5)	7.5 (2.0)	24.5 (2.3)	7.8 (1.6)	6.5 (2.5)	3.9 (2.0)	20.8 (4.6)
Multiplicity	21	3	18	4	19	6	27
Completeness (%)	99.9	91.7	99.9	99.5	99.1	99.1	99.9
No. mol. /asymm. unit	2	16	2	8	2	8	2
Refinement							
R _{work} /R _{free} (%)		22.9/25.3	20.0/24.5	19.6/24.7	23.3/27.1	23.6/27.9	25.2/27.9
No. atoms		52441	6327	23482	6169	23618	5179
Protein		51136	6275	23275	6136	23414	5144
Ligand/ion		/	43	204	18	204	/
Water		1305	9	3	15	/	35
Average B-factor (Å ²)							
Protein		58.1	84.6	84.1	165.3	140.3	85.3
Ligand/ion		/	122.6	92.6	188.3	155.0	/
Water		47.2	60.4	39.8	137.3	/	81.3
r.m.s.d.							
Bond lengths (Å)		0.002	0.005	0.002	0.003	0.002	0.002
Angles (°)		0.484	0.72	0.50	0.57	0.45	0.49
Ramachandran							
Favored (%)		95.6	95.5	94.3	94.4	94.5	94.6
Allowed (%)		4.3	4.6	5.1	5.3	5.3	5.2
Outliers (%)		0.1	0.0	0.1	0.4	0.3	0.2

The D172N-deltaN30-IMP is a bit more confusing as the CC1/2 looks OK and but the Rmeas is very high. What do the authors think is the reason for this and what have they done to improve these statistics?

We entirely agree with the referee. Nevertheless, since the D172N-deltaN30 IMP bound structure had a valid CC1/2 value we decided not to touch this structure. As concerns Rmeas, at this level whatever we do when trying to improve this parameter (cutting data according to the signal to noise ratio, resolution, only use the "first" part of the collected data in order to reduce errors due to radiation damage...) it remains this high and this seems to be inherent to the system. We believe that one of the reasons is that there is a slight "fall-off" in intensity

around approximately 6Å. We have managed to collect two other data set on this construct and the Rmeas was even higher. It should be mentioned that it has been very difficult to obtain crystals of this truncated enzyme. The main aim with these data was to show the role of the N-terminal in this enzyme, and thereby the overall conformation of this construct – there is absolutely no doubt about this when using these data even though being relatively low resolution. The $I/\sigma(I)$ is 1.4 in the outermost resolution shell. We can even add that despite the low resolution and the limited quality of the data we can see electron density around voluminous residues. The structure of this construct (together with the other structures in this manuscript/study) is essential for the validation of our hypotheses, and for that reason we decided to include it.

In my view, this should be addressed, as the resolution of the data must be appropriate for the biological conclusions that are drawn. The placement of the secondary structural elements will no doubt be right at these higher resolutions, making description of conformational changes reliable, but the side chain interactions described are less reliable at these interactions. Perhaps the authors should also give electron density maps for key regions in these lower resolution maps?

We of course understand and fully agree with the referee that the resolution of the data must be appropriate for the drawn biological conclusions. As suggested by the reviewer we have made a figure showing 2Fo-Fc electron density maps contoured at 1.0 σ in order to show that the data indeed are reliable for identifying side-chain interactions in key regions (see hereafter).

We have not included them in the manuscript (would be in the Supplementary part if added), but if this referee and the editor judges that they should be included we will of course do so. It also should be mentioned that when superimposing structures issued from the newly reprocessed data onto structures from the earlier processed data, we do not see any significant differences.

Reviewer #2 (Remarks to the Author):

The revised manuscript by Carrique L et al. is much improved and addressed the main point that was raised by both referees and the editor (regarding the lack of data to support the physiological relevance of PfISN1). Most of the other points were also addressed in the revised manuscript.

We are happy to hear that this reviewer also finds our manuscript much improved.

Several mediocre data presentations/figures in the supplements still distract from the interesting data and some minor errors and imprecisions still remain to be corrected before publication. For example:

Throughout the whole manuscript and supplements: "The unit of v is μmol of product formed in one sec." The statement is wrong and should be corrected. The y-axis of the Michaelis-Menten plots now shows " $v/[\text{enzyme}]$ " with the correct unit " sec^{-1} " with v as a change of concentration (not an amount of substance!) over time (unit " $\text{M}/(\text{M} \times \text{sec}) = \text{sec}^{-1}$ "). The authors apparently mix up the definition of an enzyme unit (conversion of μmol substrate per min) and the definition of a reaction velocity. It is interesting to note that 11 authors publish a manuscript with half of its data on enzyme kinetics and none of the authors recognizes this very basic error in the legends of Fig. 3, Fig. S6, Fig. S7, Fig. S11 as well as the methods section. Or was there a general mistake regarding the data analysis?

There is no general mistake in data analysis. We apologize for this oversight and thank the reviewer for the very careful reading. These statements has now been corrected accordingly in legends to Figs 3, S6, S7 and S11 as well as in the methods section to " v , the initial rate is the change in concentration of the product over time".

Supplementary Fig.1: "B)List of all Plasmodium species containing ISN1 genes." Should be "List of all currently sequenced Plasmodium species containing ..."

Okay, this has now been done

Supplementary Fig. 3: Please do not use the term "weight" in combination with the unit kDa. Use the term "mass" instead (3 times in the legend).

Yes, thanks for pointing out and sorry for that. This has now been done together with correction of the same in the figure itself. The incorrect term "weight" appears twice in Supplementary Fig. 2 also and we have made the corrections here as well.

Supplementary Fig. 6: The insets are hardly readable. Please improve the quality of the figure according to the standards of the journal.

Yes, sorry for that. This has now been improved in order to meet the standards of the journal. Should they still be too difficult to read (due to reformatting), the original figures are available.

Supplementary Fig. 9: Kd in panel B should be apparent Kd.

Yes, sorry for that. This has now been changed.

Supplementary Fig. 10: The figure includes blurry screen shots and a small inset. Please improve the quality of the figure according to the standards of the journal.

Yes, sorry for that. This has now been done.

Supplementary Fig. 11: There are confusing background numbers in panel B (which is not labelled) that might have been added manually. Please correct the figure.

Okay, this turned out to be a formatting problem (corresponding to the line numbers) when converting the word file to a pdf file, and which we couldn't see in the original file. The format has now been changed and this should not happen again. Labels have been added.

Supplementary Fig. 14: The data fit is still poor with a systematic distribution of data points above (ratio 0.3 to 0.7) and below (ratio 0.6 to 1.6) the fit. A likely explanation for the observed discrepancy is the selection of a static single site binding model. Even though the stoichiometry of 1:1 might not change with better models/fits that take into account a potential cooperativity of the oligomeric protein, the authors appear to miss a good opportunity to further support their statement on effector-dependent domain rearrangements that are relevant for catalysis (e.g., IMP-induced fit of a closed conformation).

The ITC data was fitted to different models using "ORIGIN" which is available with Microcal. The models that we have fitted it to are one-site binding and sequential one to four site binding. Fitting to two site binding yielded an error as the model does not converge on iteration. Statistically, the fits to three and four sequential binding are better than a simple model for just one site binding. However the values of ΔH and ΔS vary from negative to positive across the binding sites. Finally the K values don't show a sequential increase or decrease in binding affinity.

We did this experiment to establish number of binding sites. Since the structure showed variation in the N-terminus, half site binding may have been possible. However the stoichiometry was 1:1.

To summarize, though there is a statistical improvement in the fit to three and four sequential binding, the values of the parameters derived do not make physical sense. For this reason we cannot see how we can put forward a solid argument without over interpreting.

Page 6, line 122: At a concentration of 10 mM, only IMP and, to a much lower extent, AMP and p-nitrophenyl phosphate (pNPP, non-physiological substrate) were found to be substrates with specific activity values of 3.2 ± 0.5 , 0.07 ± 0.03 and 0.002 ± 0.0002 sec⁻¹, respectively. (I tried to correct multiple errors in this sentence and hope that it makes more sense now).

We thank the referee for the effort and have now replaced the former sentence.

Page 6, line 128: "whereas the Km value dropped" should be "apparent Km".

Okay thanks – corrected.

Page 7, line 131: " drop in Km suggests" should be "apparent Km".

Okay thanks – corrected.

Page 7, line 133: " The pK1 and pK2 values of 4.95 ... reflects that both substrate ionization and an acid group on the enzyme are involved in binding " should be " The apparent pK1 and pK2 values of 4.95 ... might reflect that both substrate ionization and an acid group of the enzyme are involved in binding." If the authors insist on their wording they would have to show controls with $\log(K_{cat(app)}/K_m(app))$ plots for mutants and variable charged substrate analogues to support the statement.

Okay thank you. We have changed this accordingly.

Page 10, line 201: " higher catalytic efficiency (kcat/Km)" should be "higher catalytic efficiency (kcat(app)/Km(app))".

Thank you for noting, this is now corrected.

Legend Fig. 2: " orientation as seen in figures 3C and 3D are indicated." Should be "2C and 2D"

Yes indeed, thanks for noting, this is now corrected.

Legend Fig. 2: " behind subunits b (left-hand side) and a (right-hand" ... " an NTRD from subunit a" ... " two OD's of subunits β and d." Please use greek symbols for all subunits.

Correct, this has escaped our attention. Thanks for noting, this is now corrected.

Abstract and Discussion: "With earlier evidence on isn1 upregulation in female gametocytes, the structures reported in this study form the basis for new transmission-blocking agents." and "The presence of ISN1 in the plasmodial species causing malaria in humans and its absence in the host raises the possibility of developing therapeutics targeted through this enzyme." This referee still thinks that the therapeutic aspect is rather far-fetched but leaves it up to the editor whether the two statements make sense or not.

*Upon request from this referee but also from the editor, we have now reformulated the former statement in the abstract from "With earlier evidence on isn1 upregulation in female gametocytes, the structures reported in this study form the basis for new transmission-blocking agents." to ".....the structures reported in this study **may contribute to initiate the design for possible transmission-blocking agents.**" We believe that the sentence now has been "toned down" in order to emphasize the "suggestive nature" of the sentence. If this*

referee and the editor still think that this is too far-fetched we will simply take out the sentence. In the discussion section the sentence: "The presence of ISN1 in the plasmodial species causing malaria in humans and its absence in the host raises the possibility of developing therapeutics targeted through this enzyme." has been deleted.

REVIEWERS' COMMENTS:

Reviewer #2 (Remarks to the Author):

The authors have adequately addressed all remaining points.